# Plasmids manipulate bacterial behaviour through translational regulatory crosstalk

**Catriona M. A. Thompson**[1,2], **James P. J. Hall**[3], **Govind Chandra**[1], **Carlo Martins**[1], **Gerhard Saalbach**[1], **Supakan Panturat**[1], **Susannah M. Bird**[4], **Samuel Ford**[4], **Richard H. Little**[1], **Ainelen Piazza**[1], **Ellie Harrison**[4], **Robert W. Jackson**[5], **Michael A. Brockhurst**[4,6], **Jacob G. Malone**[1,2]*

**1** Department of Molecular Microbiology, John Innes Centre, Colney Lane, Norwich, United Kingdom, **2** School of Biological Sciences, University of East Anglia, Norwich Research Park, Norwich, Norfolk, United Kingdom, **3** Department of Evolution, Ecology and Behaviour Institute of Infection, Veterinary and Ecological Sciences University of Liverpool, Crown Street, Liverpool, United Kingdom, **4** Department of Animal and Plant Sciences, University of Sheffield, Sheffield, United Kingdom, **5** School of Biosciences, University of Birmingham, Edgbaston, Birmingham, United Kingdom, **6** Division of Evolution and Genomic Sciences, School of Biological Sciences, University of Manchester, Manchester, United Kingdom

* Jacob.malone@jic.ac.uk

**Data Availability Statement:** All relevant data are within the paper and its Supporting Information

## Abstract

Beyond their role in horizontal gene transfer, conjugative plasmids commonly encode homologues of bacterial regulators. Known plasmid regulator homologues have highly targeted effects upon the transcription of specific bacterial traits. Here, we characterise a plasmid translational regulator, RsmQ, capable of taking global regulatory control in *Pseudomonas fluorescens* and causing a behavioural switch from motile to sessile lifestyle. RsmQ acts as a global regulator, controlling the host proteome through direct interaction with host mRNAs and interference with the host's translational regulatory network. This mRNA interference leads to large-scale proteomic changes in metabolic genes, key regulators, and genes involved in chemotaxis, thus controlling bacterial metabolism and motility. Moreover, comparative analyses found RsmQ to be encoded on a large number of divergent plasmids isolated from multiple bacterial host taxa, suggesting the widespread importance of RsmQ for manipulating bacterial behaviour across clinical, environmental, and agricultural niches. RsmQ is a widespread plasmid global translational regulator primarily evolved for host chromosomal control to manipulate bacterial behaviour and lifestyle.

## Significance statement

Plasmids are recognised for their important role in bacterial evolution as drivers of horizontal gene transfer. Less well understood are the effects of plasmids upon bacterial behaviours by manipulating the expression of key bacterial phenotypes. Until now, examples of plasmid manipulation of their bacterial hosts were limited to highly targeted transcriptional or translational control of a few related traits. In contrast, here we describe the first plasmid global translational regulator evolved to control the bacterial behavioural switch from a motile to a sessile lifestyle and bacterial metabolism, mediated through manipulation of the bacterial proteome.

files, with the following exceptions: Processed RNA-seq data is deposited in ArrayExpress (E-MTAB-11868). Experimental data for the proteomic experiments is deposited in ProteomeXchange (PXD033640). Scripts and bioinformatic analyses can be found at www.github.com/jpjh/PLASMAN_RsmQ.

**Funding:** JGM and CMAT were supported by BBSRC Responsive mode Grant BB/R018154/1 to JGM. JGM and RHL were supported by BBSRC Institute Strategic Programme Grant BBS/E/J/000PR9797 to the John Innes Centre. SP was supported by a Royal Thai Government PhD Scholarship. AP was supported by UKRI-BBSRC Grant BB/T004363/1 to JGM. JPH was supported by BB/R014884/1. MAB, SF and SMB were supported by BBSRC grants BB/R014884/1, BB/R014884/2, BB/R018154/1 and NERC grants NE/R008825/1, NE/R008825/2. EH is supported by a NERC Independent Research Fellowship NE/P017584/1. RWJ is supported by BBSRC grants BB/R014884/1 and BB/T010568/1. The funders had no role in study design, data collection and analysis, decision to publish, or preparation of the manuscript.

**Competing interests:** The authors have declared that no competing interests exist.

**Abbreviations:** dsDNA, double-stranded DNA; HMM, hidden Markov model; KB, Kings broth; LB, Lysogeny broth; ncRNA, noncoding RNA; NMDS, non-metric multidimensional scaling; PCC, plasmid-chromosome crosstalk; RBS, ribosome-binding site; RTS, real time search; SDC, sodium deoxycholate; SPR, surface plasmon resonance; SPS, synchronous precursor selection; ssDNA, single-stranded DNA; TCS, two-component system; T6SS, type VI secretion system; WT, wild-type.

Moreover, this global translational regulator is common across divergent plasmids in a wide range of bacterial host taxa, suggesting that plasmids may commonly control bacterial lifestyle in the clinic, agricultural fields, and beyond.

## Introduction

Bacteria regulate the expression of functional traits in response to their environment, enabling colonisation of diverse ecological niches. However, control over bacterial gene regulation is not exclusively under the control of the bacterial genome [1,2]. The mobile genetic elements that inhabit bacterial hosts, such as conjugative plasmids, commonly encode homologues of bacterial regulators [3,4]. The introduction of plasmid-encoded regulator homologues into the bacterial cell can rewire the gene regulatory networks of the bacterium, potentially altering the expression of bacterial traits, a process termed plasmid-chromosome crosstalk (PCC, [5]). However, how and why plasmid-encoded regulators would manipulate the expression of bacterial traits is poorly understood.

To date, well-characterised PCCs involve plasmid-encoded transcriptional regulators that alter the expression of specific bacterial traits. For example, in *Acinetobacter baumannii* several multidrug resistance plasmids encode transcriptional repressors of the bacterial type VI secretion system (T6SS) [6]. Plasmid-mediated repression of the T6SS enhances plasmid horizontal transmission by ensuring that plasmid recipient cells are not killed by the donor's T6SS apparatus [7]. Similarly, plasmid-encoded transcriptional regulators alter the expression of several chromosomal regulators of virulence associated traits in *Rhodococcus equi*, thus enhancing survival of both the bacterium and the plasmid in macrophages by stalling phagosomal maturation [8]. Together, these examples suggest that plasmid-encoded regulatory homologues may have important fitness consequences for the plasmid, either through horizontal replication, through conjugation to new host cells or through vertical replication within the current host cell [3].

The molecular mechanisms of known PCC involve plasmid-encoded transcriptional regulators causing targeted changes to the expression of small numbers of chromosomal genes. Although transcriptional regulation is important for bacterial survival and adaptation, bacteria also rely on translational regulation to respond to changes in their environment [9]. Bacteria are able to exert this control by deploying second messenger signals [10], directly altering the ribosome [11] or impacting mRNA stability and accessibility via pathways such as Gac-Rsm [12,13]. Although specific targeted instances of translational control have been observed [14,15], it is currently unknown whether conjugative plasmids are able to manipulate global translational regulatory pathways.

The Gac-Rsm pathway is one of the best characterised translational regulatory systems in pseudomonads [16–20] and controls a wide variety of traits including biofilm formation [21], motility [22], quorum sensing [23], siderophore production [24], and virulence [12,25]. Gac-Rsm is highly conserved within the *Pseudomonas* genus and comparable systems exist in a wide range of bacteria [12,20,24,26,27]. Rsm proteins are able to interact directly with the bases AnGGA around the ribosome-binding site (RBS) of their target transcript [27–30]. Rsm proteins can both activate and repress bound mRNA transcripts, either by opening up the mRNA to allow ribosomal access to the RBS, or by making the RBS inaccessible [30–32]. This allows Rsm proteins to exert tight translational control over a wide range of targets to impact bacterial phenotypes [33]. The activation of Rsm proteins are regulated by the activation of the GacA/S two-component system (TCS), which in turn is activated by a complex, but largely uncharacterised, set of environmental cues. Upon activation, GacA promotes transcription of the small-regulatory RNAs RsmY and RsmZ, which leads to the sequestering of regulatory

Rsm proteins away from their mRNA targets through competition for binding [30,34]. The number of Rsm proteins encoded by individual *Pseudomonas* species varies, with each protein having both unique and overlapping regulons with other Rsm proteins [35]. The large number of traits regulated by the Gac-Rsm system suggests that there could be significant effects on bacterial behaviour caused by PCC manipulating this system.

In this study, we investigate the role of translational regulation in mediating PCC between *Pseudomonas fluorescens* SBW25 and the 425 kb conjugative plasmid pQBR103. *P. fluorescens* is a common, soil-dwelling, plant growth-promoting bacterium that is capable of accepting diverse plasmids, including those from the pQBR family of large conjugative plasmids [36,37]. Both SBW25 and the pQBR plasmids were first isolated in the 1990s from the sugar beet rhizosphere at Wytham Woods in the United Kingdom [36,38]. The ability of several of the pQBR plasmids to persist within *P. fluorescens* strains across a range of environments including in soil, on plants, and in lab media has been well documented [36,37,39,40]. Moreover, acquisition of pQBR103 by *P. fluorescens* SBW25 alters the expression of approximately 440 chromosomal genes [40,41]. The large-scale regulatory disruption caused by pQBR103 can be negated by a range of compensatory mutations restoring wild-type (WT) expression levels, including loss-of-function mutations affecting the bacterial TCS *gacA/S*. Notably, while the genetic sequence of pQBR103 encodes a range of accessory functions including mercury resistance and UV resistance, it also encodes a homologue of the widespread *rsmA* bacterial translational regulator gene, which we identify here as *rsmQ*.

To understand the function of *rsmQ*, we explored the distribution of *rsm* genes on plasmids, and the effects of *rsmQ* on the transcriptome and proteome of SBW25, as well as on the expression of key bacterial ecological traits. Further, we biochemically characterised the interactions of RsmQ with a close structural proxy for RNA (single-stranded DNA (ssDNA)) and with the bacterial Rsm proteins. Our findings show that *rsm* genes are widespread on *Pseudomonas* plasmids and that RsmQ interacts with the resident Gac-Rsm system and the host RNA pool, binding to specific nucleotide motifs in order to post transcriptionally regulate translation. RsmQ extensively remodelled the SBW25 proteome including altering production of metabolites nutrient uptake and chemotaxis pathways, despite having only a limited impact on the SBW25 transcriptome. In turn, RsmQ translational regulation altered the expression of ecologically important bacterial behaviours, most notably motility, conjugation rate, and carbon source metabolism. Together, our findings expand the known molecular mechanisms causing PCC to include translational regulator homologues, which act in this case to extensively manipulate bacterial behaviour by altering the expression of a range of ecologically important bacterial traits. These findings have broad implications for understanding the role of plasmids in microbial communities.

## Results

### Plasmids encode regulatory protein homologues

The ORF *PQBR443*, hereafter *rsmQ*, was identified on pQBR103 as a homologue of the chromosomal *csrA/rsmA* genes found widely within proteobacteria. We hypothesised that this gene could act as a mediator of PCC [4]. To identify whether carriage of an *rsm* homologue is peculiar to pQBR103 or is a general phenomenon across plasmids, we investigated the distribution of *rsmQ* homologues in the 12,084 plasmids of the COMPASS database [42]. Within this set, and consistent with previous studies [43], we detected 106 putative *rsmQ* homologues on 98 plasmids (0.8%), mostly isolated (92/98) from proteobacteria, particularly Pseudomonadaceae and Legionellaceae (Figs 1A and S1C). The distribution of *rsm*-containing plasmids was not uniform across taxa (Fisher's exact test, $p < 0.0005$: approximately 20% of Pseudomonadaceae

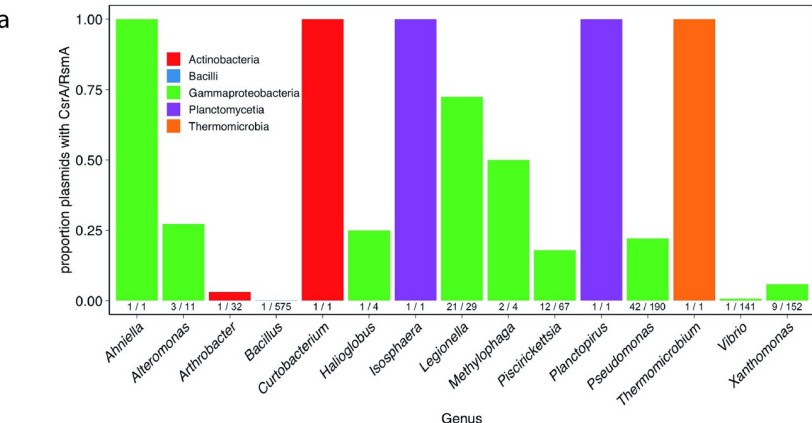

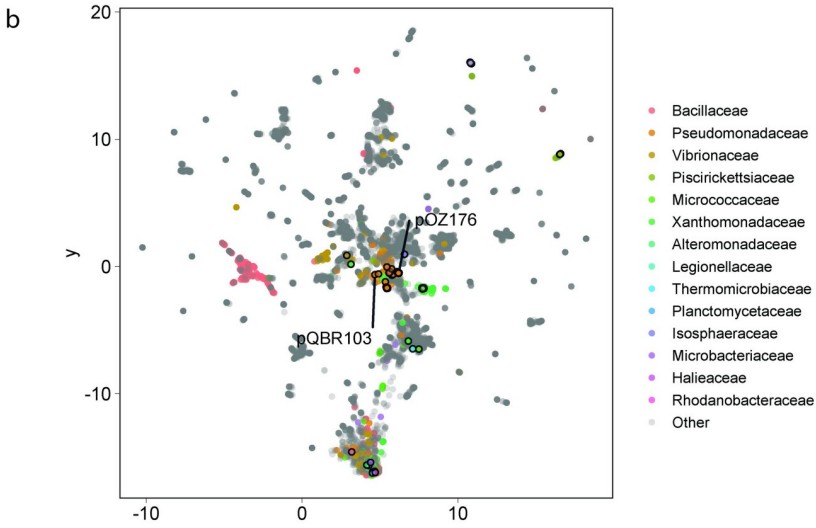

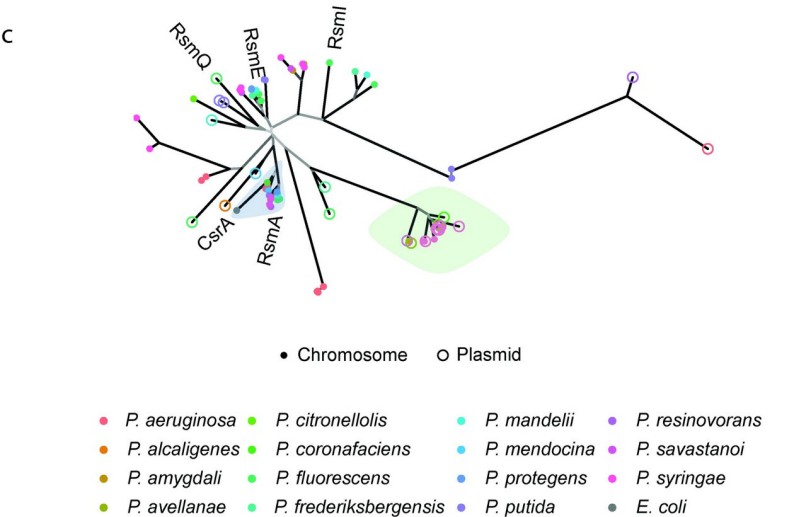

**Fig 1. RsmQ is found on a wide range of conjugative plasmids.** (a) Taxonomic distribution of plasmid borne *csrA/rsmA* homologues identified in COMPASS. (b) COMPASS plasmid diversity represented by NMDS of MASH sequence distances. Families with ≥1 plasmid with a *csrA/rsmA* homologue are coloured according to the legend. Plasmids encoding *csrA/rsmA* homologues are outlined in black. Selected plasmids from various taxa are annotated. (c) Unrooted phylogenetic tree of *Pseudomonas csrA/rsmA* homologues from COMPASS, with corresponding chromosomal homologues (where available) and genes from selected reference strains. Branches leading to nodes with >80% bootstrap support are coloured black, with decreasing support indicated with increasingly pale grey. *P. fluorescens* SBW25 *csrA/rsmA* homologues are labelled, as is pQBR103 *rsmQ*, and *E. coli csrA*. The blue highlight indicates a well-supported (bootstrap support 0.84) group of *rsmA*-like homologues. The green highlight indicates the group of related plasmid and chromosomal genes from plant pathogen *Pseudomonas* discussed in the text. All data and analyses are available on github (PLASMAN_RsmQ) with data taken from the COMPASS database. NMDS, non-metric multidimensional scaling.

(41/196) and Piscirickettsiaceae (12/67) plasmids, and over 50% of Legionellaceae (21/29) plasmids contained *rsm* homologues, while no *rsm* homologues were detected on any of the 3,621 Enterobacteriaceae plasmids. *rsm*-containing plasmids were relatively large, with the smallest at 32.4 kb, sitting at the larger end of the size distribution for each taxon (S1A Fig). There was no general association between *rsm*-carriage and plasmid mobility across taxa, although within Legionellaceae, *rsm*-encoding plasmids tended to be conjugative (Fisher's exact test Bonferroni-adjusted *p*-value = 0.02). Within Pseudomonadaceae, *rsm*-encoding plasmids tended to have proportionally more genes with predicted *rsm* binding sites (Kolmogorov–Smirnov test, $p = 0.012$, S1B Fig). Overall, these patterns suggest that plasmid carriage of *rsm* is not uncommon, but is taxon specific, indicating a functional role that is associated with particular groups of microorganisms.

It is possible that *rsm*-encoding plasmids have recently spread horizontally across different species. If this was the case, we would expect *rsm*-encoding plasmids to be more similar to one another than to non-*rsm*-encoding plasmids within each taxon. To investigate the diversity of *rsm*-encoding plasmids relative to the other plasmids in COMPASS, we performed UMAP non-metric multidimensional scaling (NMDS) on pairwise MASH distances between plasmids [44–46]. Within the diversity of plasmids in COMPASS, *rsm*-containing plasmids are diverse and often cluster close to non-*rsm*-containing plasmids isolated from the same taxa (Fig 1B), suggesting that carriage of *rsm* regulators by plasmids is a convergent trait that has emerged several times over.

Global regulatory genes may be frequently (re)acquired by plasmids from the bacterial chromosome. Alternatively, these genes may have a prolonged association with plasmids and evolve distinctly to chromosomal genes. To investigate these possibilities, we built a phylogenetic tree of the *csrA/rsmA* homologues from all *Pseudomonas* plasmids and their associated chromosomes (where available), alongside the *rsm* genes from 7 diverse *Pseudomonas* strains: *Pseudomonas protegens* CHA0, *P. fluorescens* Pf0-1, *P. protegens* Pf-5, *P. fluorescens* SBW25, *Pseudomonas putida* KT2440, *Pseudomonas aeruginosa* PAO1, and *P. aeruginosa* PA14 (Fig 1C). Chromosomal homologues of *csrA/rsmA* formed several distinct clusters (bootstrap support >80%), with 1 cluster including *P. fluorescens* SBW25 *rsmA* and the *Escherichia coli* homologue *csrA*. However, plasmid-borne *csrA/rsmA* homologues were more divergent than those that were chromosomally encoded (Fig 1C). Additionally, chromosomal homologues (including the *P. fluorescens* SBW25 genes *rsmE* and *rsmI*) formed a distinct cluster. Consistent with the phylogenetic analysis, sequence variation among chromosomal *rsm* homologues was significantly lower than when comparing chromosomal- with plasmid-borne *rsm* homologues (Wilcoxon test, Bonferroni-adjusted $p < 0.0001$) or when comparing plasmid-borne *rsm* homologues with one another (Wilcoxon test, $p < 0.0001$). The principal exception to this pattern was a cluster of closely related plasmid and chromosomal *rsm* genes from plant pathogenic *Pseudomonas* (green highlighted, Fig 1C). However, it is possible that some of these

chromosomal variants are associated with chromosomally located mobile genetic elements, as at least one of these homologues is located on an annotated integrative conjugative element [47].

Overall, our comparative sequence analysis suggests that diverse plasmids have independently acquired *rsm* homologues, which then evolve and diversify as part of the plasmid mobile gene pool, distinct from their chromosomal counterparts. Although plasmid-encoded *rsm* homologues are widespread among plasmids [43], very little is currently known about their role in PCC or how they might impact bacterial behaviour.

## RsmQ binds to specific RNA targets

Despite a high degree of sequence similarity, it was unknown if RsmQ would be functionally similar to the chromosomally encoded SBW25 Rsm proteins (RsmA/E/I). Rsm proteins from *Pseudomonas* species interact with a conserved RNA sequence (AnGGA), with these bases interacting directly with the proteins' conserved binding site (VHRE/D) [29,30]. To confirm whether RsmQ acts similarly, we designed a high-throughput method to examine the nucleotide binding properties of RsmQ in vitro using the ReDCaT surface plasmon resonance (SPR) system [48], which is primarily designed for examining double-stranded DNA (dsDNA)–protein interactions. Because Rsm proteins only interact with the nucleotide bases of RNA molecules, protein–nucleic acid interactions can be effectively examined using ssDNA probes. ssDNA probes containing the predicted RNA target sequence (ACGGA) and a nonspecific scrambled sequence were synthesised with the ReDCaT linker at the 3′ end, with either a linear or a hairpin secondary structure with the potential binding site at the top of the hairpin.

RsmQ interacted strongly with both the minimal (GGA) and full length (ACGGA) binding sites when these were at the top of a hairpin loop. When the binding sequence was presented in a linear oligo RsmQ could interact but quickly dissociated, suggesting that the preferred binding site is open at the top of a hairpin loop (S2A Fig). No interaction was seen between RsmQ and a scrambled binding site confirming that the binding is specific to the target GGA/ACGGA sequence (Fig 2).

Next, we tested if the RNA binding interaction was co-ordinated by the conserved VHRD/E motif at the C-terminus of RsmQ by examining the binding of 2 RsmQ mutants, in which key residues within the motif were changed to alanine residues (H43A and R44A), to the ssDNA probes. The alteration of these residues significantly reduced the efficiency of RsmQ binding to the target sequence (Fig 2A). Finally, to confirm the minimum RNA-binding sequence, a series of near-identical ssDNA nucleotides were tested containing the simple and full binding site sequences with a single base pair change in each case. RsmQ preferentially bound to the known binding sites GGA and A(N)GGA with a markedly higher affinity than to any of the alternate sequences tested and with a slight preference for ATGGA/AGGGA sequences, further supporting the hypothesis that RsmQ is a specific RNA binding protein that functions similarly to the chromosomal Rsm proteins (Fig 2B). Previous work by Duss and colleagues [49] has shown that RsmE is able to directly interact with the GGA motif of the AnGGA binding motif. Therefore, to directly compare RsmQ to its chromosomal counterparts in this assay, the interaction of RsmE to each of these oligos was also examined. Interestingly, whilst RsmQ showed an affinity for all AnGGA binding sites, as well as the simplified GGA binding site, RsmE was only able to effectively interact with the ATGGA and AGGGA motifs (S2B and S2C Fig), suggesting that RsmQ may have a stronger affinity than the chromosomal Rsm proteins for their mRNA targets.

## RsmQ post transcriptionally regulates the abundance of metabolism, nutrient transport, and chemotaxis proteins

To examine the impact of RsmQ on SBW25 regulation, a plasmid lacking *rsmQ* was created by allelic exchange in a kanamycin resistance gene-containing variant of pQBR103 (pQBR103^Km)

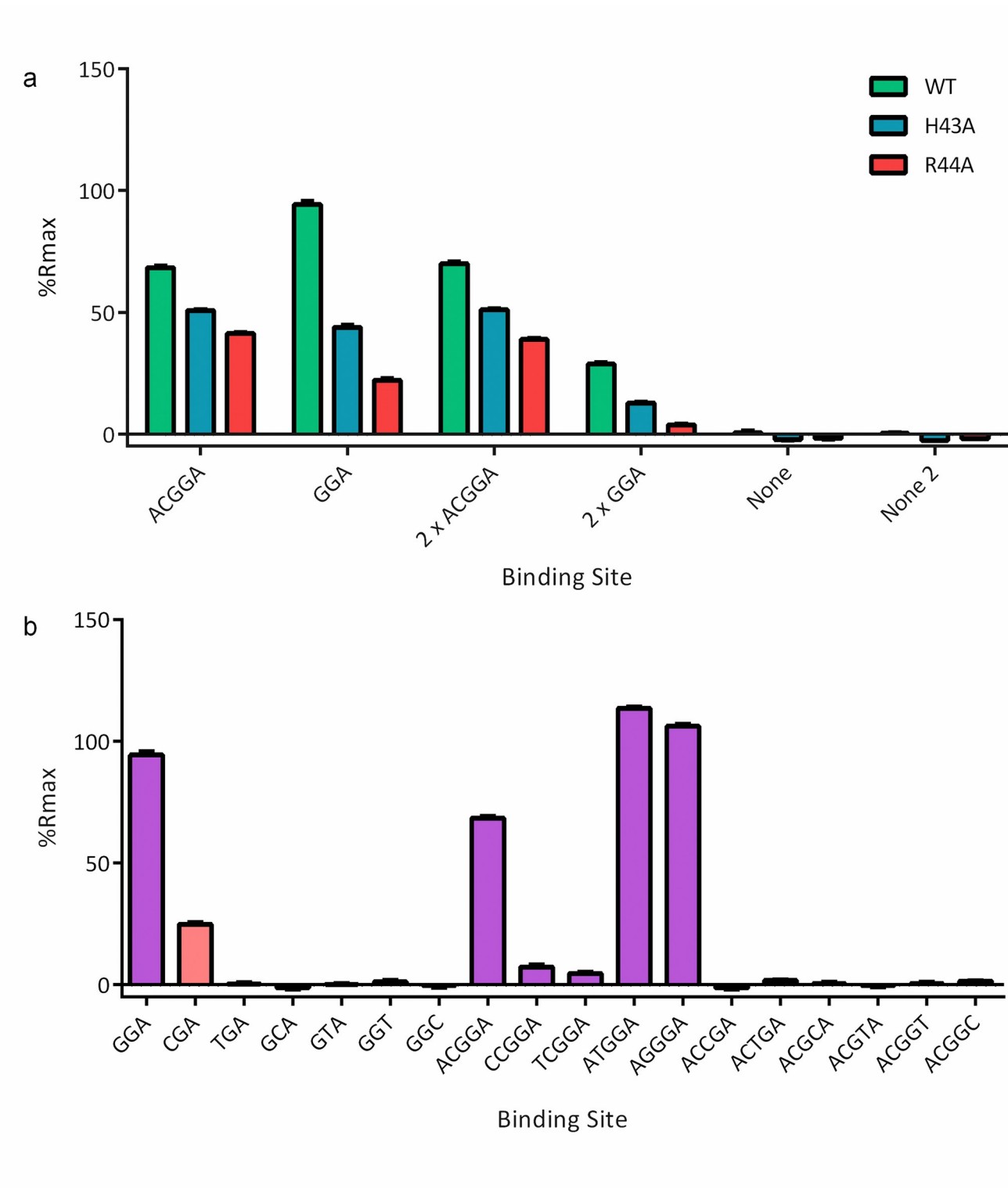

**Fig 2. RsmQ interacts with its preferred binding sequence (GGA/AnGGA) and this interaction is mediated by the VHRE/D binding site.** (a) Percentage $R_{max}$ values for RsmQ WT (green) H43A (blue) and R44A (red) binding to ssDNA containing the binding sites shown above. Two-way ANOVA showed an effect of both binding site ($p < 0.0001$) and mutation ($p < 0.0001$) on $R_{max}$. (b) Percentage $R_{max}$ values for WT RsmQ binding to ssDNAs containing the above binding site. Binding sites that are predicted to bind RsmQ are shown in purple. All oligos are designed as hairpins and results shown are for RsmQ at a concentration of 100 nM. Error bars represent the standard deviation of 2 replicates. Data are available in S1 Data. ssDNA, single-stranded DNA; WT, wild-type.

and freshly conjugated into SBW25. Expression of the chromosomal *rsm* genes generally peaks as growing cells make the switch from exponential to stationary phase growth [50]. Therefore, *P. fluorescens* SBW25 cells carrying pQBR103[Km] or pQBR103[Km]-Δ*rsmQ* or no plasmid were grown to late exponential phase ($OD_{600}$ = 1.4) in KB liquid medium and a combination of RNA-seq and TMT quantitative proteomics were used to examine changes in mRNA and protein abundances. Five independent transconjugants for each plasmid as well as 5 independent colonies of the plasmid-free strain were used for these experiments.

Contrary to previous studies of exponentially growing cultures where carriage of pQBR103 caused large-scale transcriptional changes [4,41,51], we observed only modest transcriptional changes associated with plasmid carriage under our growth conditions, with only 54 genes significantly up-regulated more than 2-fold and 33 down-regulated by pQBR103[Km] carriage, and similar numbers seen for pQBR103[Km]-Δ*rsmQ* (S3 Fig). Genes whose transcription was up-regulated predominantly encoded transport-related membrane proteins, such as ABC and glutamine high-affinity transporter components. In addition, *PFLU1887* and *PFLU1888* encoding components of a putative transposase were up-regulated by pQBR103[Km] carriage, consistent with previous studies [51]. Genes down-regulated by plasmid acquisition included several cytochrome C oxidases and metabolic enzymes such as L-lactate and shikimate dehydrogenases. pQBR103[Km] and pQBR103[Km]-Δ*rsmQ* had highly similar transcriptional effects, suggesting that that RsmQ had little impact on the transcription of either chromosomal or plasmid genes (S3C Fig).

In contrast, we observed major differences between the proteomes of SBW25 +pQBR103[Km] and SBW25 +pQBR103[Km]-Δ*rsmQ*, confirming that RsmQ is indeed a post transcriptionally acting translational regulator. Specifically, 581 SBW25 proteins showed at least 2-fold increased abundance in the absence of *rsmQ* (i.e., their translation is suppressed by RsmQ) and 203 showed at least 2-fold decreased abundance (Fig 3A and S1 Table). Intriguingly, RsmQ regulation predominantly affected the host proteome, with the abundances of only a small fraction of plasmid-encoded proteins (16/733) altered by the presence of *rsmQ*.

We next determined the COG functional categories of SBW25 proteins whose abundances were altered by the presence of RsmQ (Fig 3B). The 581 proteins down-regulated by RsmQ were disproportionately associated with amino acid, coenzyme, and carbohydrate transport and metabolism, as well as proteins involved in mRNA translation and ribosome stability. Among the 50 most strongly down-regulated proteins, we observed multiple inorganic ion transporters and receptor proteins, for example, pyoverdine receptors and iron transporters (PFLU3378, PFLU2545 (FpvA), PFLU0295) [52–55], a copper transport outer-membrane porin (PFLU0595 (OprC homologue)) and the PhoU phosphate ABC transporter known to repress the Pho operon (PFLU6044 [56]), alongside proteins involved in amino acid transport (e.g., ABC transporters PFLU0827 and PFLU0332 and metabolism (e.g., GlyA–PFLU565)). Together, these data suggest a role for RsmQ in the control of SBW25 nutrient acquisition and metabolism.

Conversely, proteins up-regulated by RsmQ included a large number of motility proteins and DNA recombination and repair systems, alongside a larger fraction of uncharacterised proteins than in the RsmQ down-regulated group of proteins. The most strongly up-regulated proteins included a striking number of chemotaxis pathway components. In addition to CheA (PFLU4414), we identified 5 putative methyl-accepting chemotaxis proteins (e.g., PFLU2358, PFLU3427, and PFLU2486). RsmQ up-regulated proteins also included the master-regulator of motility FleQ (PFLU4443 [57,58]) and an uncharacterised RpiR family transcriptional regulator (PFLU257). This suggests that RsmQ modifies bacterial motility through altering cellular perception of the environment and the availability of local nutrient sources, as well as by directly controlling production of motility apparatus. Interestingly, the Gac-Rsm TCS

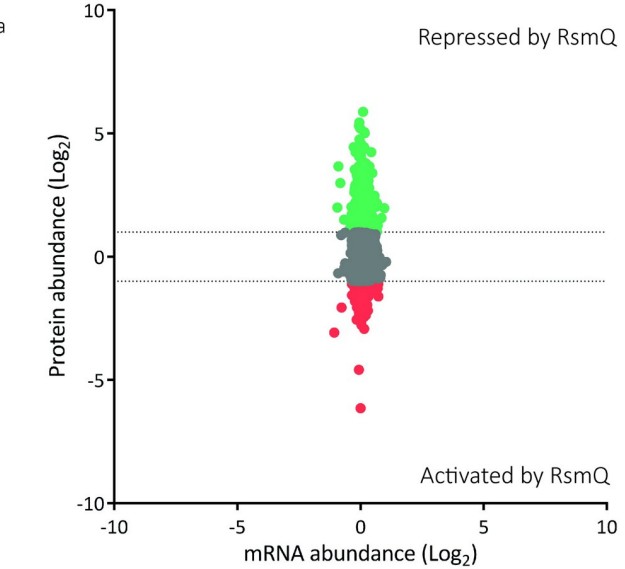

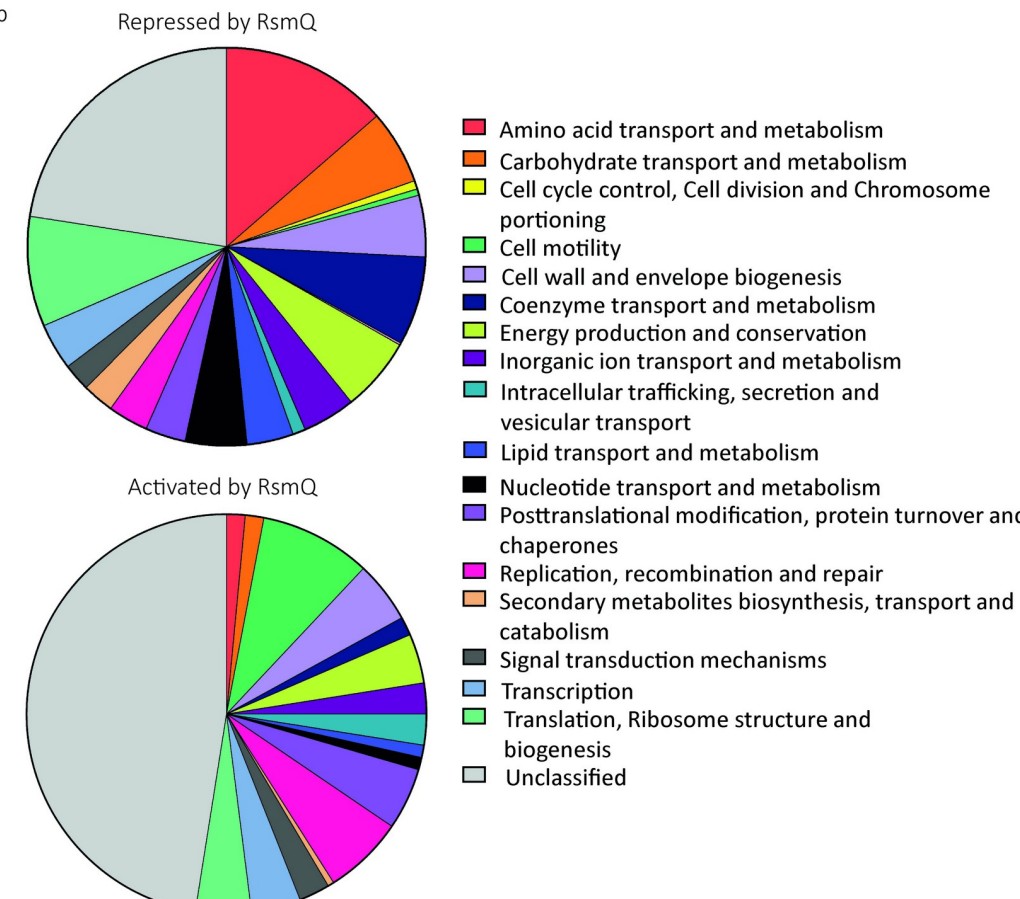

**Fig 3. The loss of RsmQ causes widescale proteomic changes.** (a) Comparative scatter plot comparing $\log_2$-fold mRNA abundance changes from RNAseq ($n = 5$) to protein abundance changes seen by TMT quantitative proteomics ($n = 3$). (b) COG categories of proteins that showed a greater than $\log_2$-fold change when *rsmQ* is lost (Repressed by RsmQ = 581, Activated by RsmQ = 203). Data are available in S1 Table and S2 Data. This data is also publicly available in ArrayExpress (E-MTAB-11868) and ProteomeXchange (PXD033640).

repressor protein RetS (PFLU0610 [59]) is also up-regulated by RsmQ, supporting a further regulatory linkage between pQBR103$^{Km}$ carriage, RsmQ function, and the Gac/Rsm pathway.

As well as genes directly involved in specific cellular processes, we also observed alterations to the expression levels of the global regulator Hfq (PFLU0520), which was shown to be down-regulated in a Δ*rsmQ* background, with corresponding shifts in a fraction of the published SBW25 Hfq regulon [60]. Intriguingly, one of the few plasmid-encoded proteins that were significantly affected by RsmQ was an Hfq homologue (pQBR0137), whose abundance increased upon *rsmQ* deletion (S1 Table). However, only the chromosomally encoded Hfq possesses a predicted RsmQ binding site.

Sequence analysis suggests that around 50% of the genes encoding proteins whose abundance is differentially regulated by RsmQ contained an AnGGA binding site upstream, or within the first 100 bp of the ORF, with an additional 25% of all genes containing the simpler GGA motif. This pattern is consistent with RsmQ binding to these mRNAs to regulate their translation. To test this, we next designed ssDNA probes to examine if RsmQ indeed targeted the binding sites of the genes predicted to be directly regulated. Binding site probes were designed to be 30 bp long with the predicted binding site in the centre of the oligo with the ReDCaT linker on the 3′ end. As previously described, SPR was performed to determine if an interaction occurred between the sequence and RsmQ in vitro. As these experiments were performed on a different occasion to previous experiments, some variation in %R$_{max}$ was observed. Therefore, the %R$_{max}$ obtained for the hairpin ACGGA synthetic binding site was used as a guide. Using this method, 5 of the potential binding site oligos showed a %R$_{max}$ of greater than 50% (Fig 4), with these being the upstream regions of PFLU0923 (ATGGA), PFLU3378 (AGGGA), PFLU1516 (AGGGA), PFLU4443 (AGGGA, FleQ), and PFLU4726 (ATGGA) with the highest binding seen with PFLU3378.

Despite an apparent preference for ATGGA/AGGGA binding sites, it is likely that the secondary structure was the overriding predictor of binding. Secondary structure predictions suggested that PFLU3378 was the only oligo with the binding site fully open at the top of the stem loop, with the rest showing partial occlusion of the ssDNA binding site by incorporation into a stem loop (S4 Fig). As the sequences tested are taken out of cellular context, these results should be considered as the minimal RsmQ-binding regulon. Nonetheless, these data confirm direct interaction between RsmQ and at least some of its predicted targets and further support the importance of mRNA secondary structure for successful RsmQ binding.

## RsmQ interacts with the host Rsm system

Notwithstanding the evidence for direct regulation of translation by RsmQ binding to mRNA, the large remainder of differentially regulated proteins without predicted Rsm binding sites suggests an indirect mechanism by which RsmQ regulates the abundance of these proteins. Given that RsmQ closely mimics the RNA binding characteristics of host Rsm proteins (Fig 2), we next investigated whether RsmQ interacts with other elements of the host Rsm regulatory pathway.

The activity of host Rsm proteins is controlled by the noncoding RNAs (ncRNAs) RsmY/Z, which act as protein sponges, sequestering Rsm proteins away from their target mRNAs [30,61]. To test RsmQ binding to the ncRNAs RsmY and RsmZ, we copied the individual stem loops of each ncRNA into ssDNA oligos of approximately 25 bp in length and attached them to the ReDCaT linker. The oligos were modelled to determine the location of the binding site in both the ncRNA and in the case of the ssDNA sequence, to determine if this was located at the top of a stem loop. Strong binding to several ssDNA probes was observed, in each case contingent on the presence of at least a GGA sequence at the top of a stem loop, with RsmY 1–25

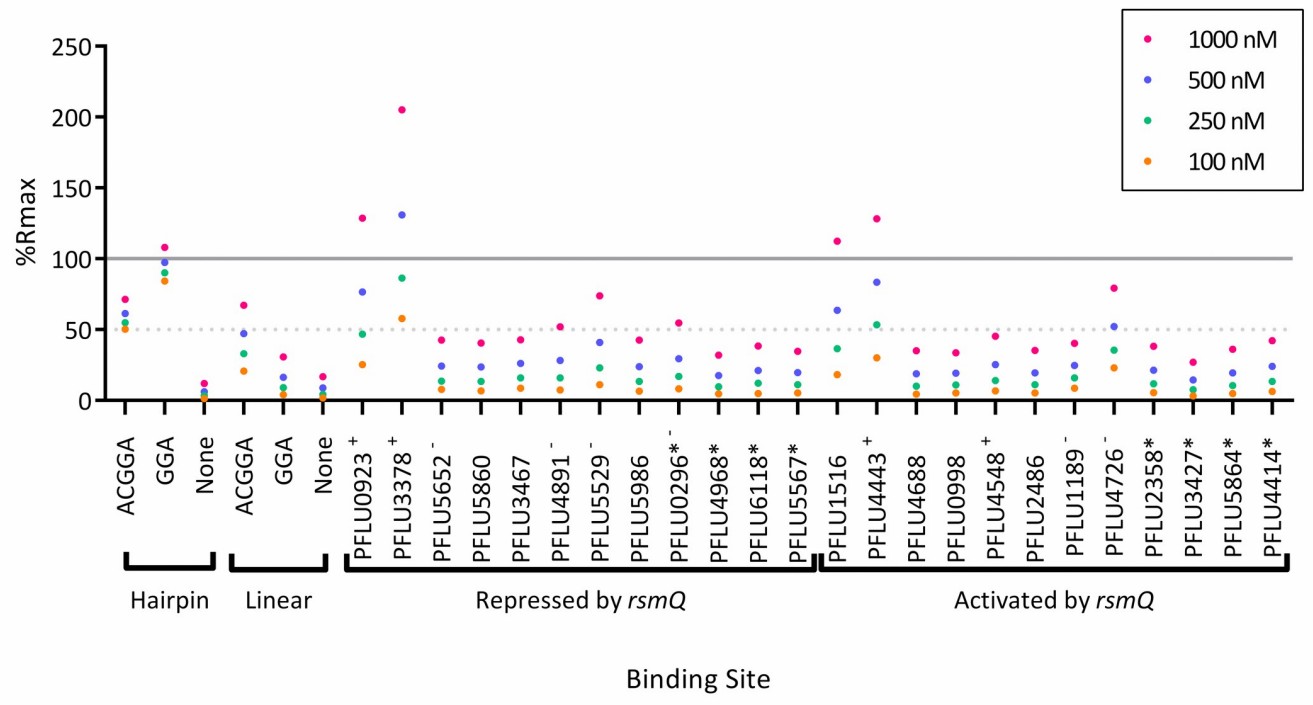

**Fig 4. RsmQ binds to the upstream regions of predicted mRNA targets.** Purified RsmQ was tested against ssDNA oligos with synthetic oligos run as a control. Oligos are labelled as in S4 Table, with the genetic identifiers used to indicate the binding sites associated with those ORFs. Genes annotated with a * indicate ORFs that contain a simplified GGA binding site but are not predicted to have a full Rsm (AnGGA) binding site within 500 bp upstream or the first 100 bp of the ORF. Oligos predicted to have the full AnGGA (+) or a partial AnGGA (-) at the open top of a stem loop are indicated. Percentage $R_{max}$ values are shown for 4 concentrations. A solid line indicates 100% $R_{max}$ and a dotted line for 50% $R_{max}$; 100% $R_{max}$ suggests 1:1 ligand protein binding as the experimentally acquired response is equal to the predicted response, with a 50% $R_{max}$ suggesting a weaker interaction or a 2:1/1:2 response. Each data point represents the mean of 2 replicates. Data are available in S3 Data. ssDNA, single-stranded DNA.

and RsmZ 26–50 having AnGGA motifs present (Fig 5A). These data suggest that RsmQ interacts with the Gac-Rsm regulatory system by binding to the host ncRNAs RsmZ and RsmY. This would lead to either an increase in RsmQ target translation as RsmQ is titrated away from its targets or an increase in RsmA/I/E binding to target mRNAs due to a reduction in available RsmZ/Y binding sites.

Rsm proteins have been seen to homodimerize and are regularly found as homodimers within the cell [27,28,62]. With the exception of RsmN from *P. aeruginosa*, they generally have a conserved tertiary structure [28]. Furthermore, an AlphaFold [63] model of RsmQ was shown to be highly similar to the crystal structures of SBW25 host Rsm proteins (Fig 5B) [62]. We hypothesised therefore that RsmQ may also interfere with regulation by forming heterodimers with host Rsm proteins. To test this, we expressed *rsmQ* and the SBW25 host *rsm* genes heterologously in *E. coli* using the Bacterial Adenylate Cyclase-Based Two-Hybrid (BACTH) system. The BATCH system allows for the screening of interacting protein partners through a cAMP cascade that is activated by bringing together the 2 adenylate cyclase domains fused to the proteins of interest [64]. Interestingly, with the exception of RsmI, we saw evidence of homo- and heterodimerisation within and between the Rsm proteins. Both RsmE and RsmQ homodimerised, and heterodimerisation was observed between all pairwise combinations of RsmA, RsmE, and RsmQ (Fig 5C and 5D). These results therefore support 2 indirect mechanisms for RsmQ regulation of the SBW25 proteome in addition to direct mRNA binding: either by sequestering ncRNAs or directly interfering with the activity of host Rsm regulators.

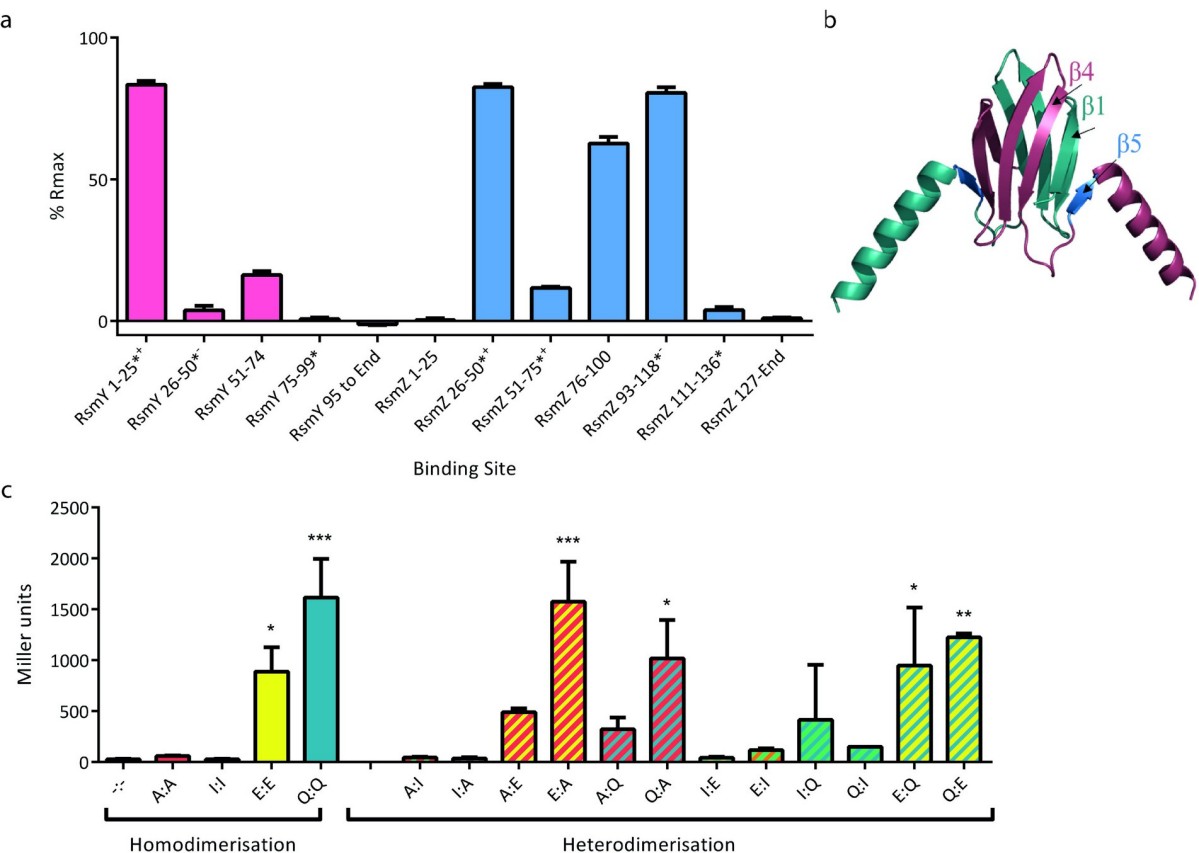

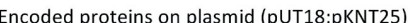

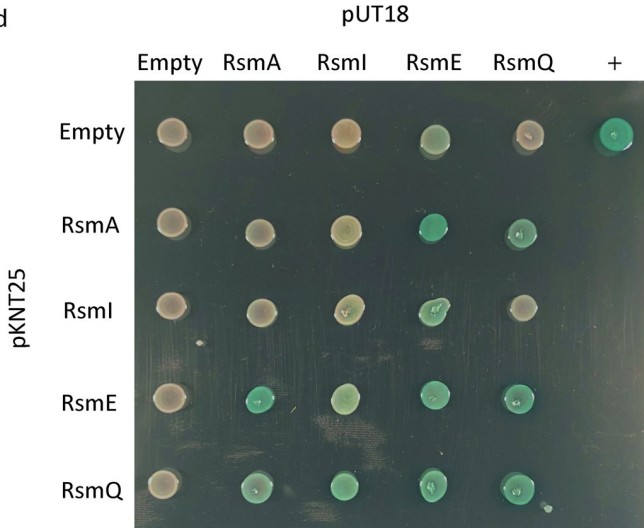

**Fig 5. RsmQ can both homo- and heterodimerise.** (a) Percentage $R_{max}$ values for RsmQ binding to portions of the ncRNAs RsmY (pink) and RsmZ (blue) showing preferential binding to ssDNAs that contained the binding site in a hairpin loop. Error bars show the standard deviation of 2 replicates and oligos containing a AnGGA binding site are indicated with a (*), with those that have the full AnGGA binding site at the top of a stem loop (+) and with the shorter GGA at the top of a stem loop (-) indicated. (b) AlphaFold model of RsmQ suggests that it forms homodimers (monomers shown in contrasting colours), with the RNA-binding region highlighted in marine (B5). (c) Quantitative

bacterial-2-hybrid β-galactosidase assays showing interactions between pUT18c and pKNT25 fusions are shown for RsmA (A), RsmE (E), RsmI (I) and RsmQ (Q). Results were analysed by a one-way ANOVA ($p < 0.0001$) with Tukey's multiple comparisons against the empty plasmid control (-:-) indicated (*$p < 0.05$, **$p < 0.01$, ***$p < 0.001$, ****$p < 0.0001$). Error bars represent the standard deviation of 2 biological replicates. Additional controls are shown in S7 Fig. (d) Representative image of qualitative β-galactosidase assays on agar plates. pKT25 fusions are shown in rows and pUT18c fusions in columns, with the indicated protein/empty vector present in each case. Data are available in S4 Data. ssDNA, single-stranded DNA.

As well as direct interaction, we considered that RsmQ may be interfering with the regulation of other chromosomally encoded Rsm proteins. To determine this, we examined the expression of each of the *rsm* genes at 3 points within the growth cycle (Mid-log, Late-log, and Early-stationary). Interestingly, although each *rsm* gene appears to be expressed at a different point within the cycle, there was no effect of plasmid carriage or *rsmQ* on expression levels (S5 Fig).

## RsmQ causes phenotypic changes in SBW25

Given the large-scale changes that RsmQ caused to the SBW25 proteome, we hypothesised that these altered protein abundances would in turn affect bacterial behaviour. To test this, we quantified the impact of RsmQ on ecologically important traits normally controlled by the Gac-Rsm regulatory system. Specifically, we initially quantified swarming motility and production of exopolysaccharide/adhesin (measured using an indirect Congo red binding assay [65]) by SBW25 in the presence and absence of *rsmQ*. To examine the direct impact of *rsmQ* on chromosomally encoded genes, *rsmQ* was expressed under an inducible promoter on a multicopy plasmid, in the absence of pQBR103$^{Km}$. Overexpression of *rsmQ* led to a complete loss of swarming motility and a significant increase in Congo red binding (Fig 6A and 6C). This suggests that *rsmQ* shifts SBW25 towards a more sessile lifestyle as characterised by reduced flagellar motility and increased production of attachment factors and/or extracellular polysaccharides associated with biofilm formation.

To test if *rsmQ* had similar effects on SBW25 behaviour when encoded on pQBR103, we repeated the experiments using SBW25 with or without pQBR103$^{Km}$ ±*rsmQ*. Acquisition of pQBR103$^{Km}$ caused reduced swarming motility and Congo red binding relative to plasmid-free SBW25. However, deletion of *rsmQ* only partially ameliorated the reduction in swarming motility (Figs 6B and S6) and had no effect on Congo red binding (Fig 6C). The expression of Rsm proteins is normally tightly controlled by the cell; these results suggest that at high concentrations RsmQ can override the native cellular control to cause drastic phenotypic changes that are not observed at the native level. This is also consistent with our proteomic data (Fig 3B), which showed little or no impact of RsmQ on the abundance of structural biofilm or motility proteins such as flagella and adhesins, but a significant impact on chemotaxis pathways.

We hypothesised that the role of RsmQ may be in the perception and uptake of specific nutrients, and therefore any phenotypic changes may be carbon source dependent. To test this, we examined the effect of the nutrient environment on swarming motility phenotypes and observed that pQBR103$^{Km}$ carriage strongly effected swarming motility on poorer carbon sources (S7 Fig), with the loss of *rsmQ* leading to a small restoration of swarming on some C-sources but not others, again suggesting *rsmQ* is involved in manipulating the cellular perception of the environment.

To examine the impact of the wider Gac-Rsm pathway on the plasmid-associated biofilm phenotype, Congo red binding was measured for plasmid-free *rsm* mutants, as well as those carrying pQBR103$^{Km}$ ±*rsmQ* (S8 Fig). Interestingly, the plasmid mediated increase in Congo red binding was maintained across all Δ*rsm* backgrounds, suggesting that no single Rsm protein is responsible for plasmid-mediated biofilm formation nor is RsmQ directly involved. Interestingly, however, we saw no significant difference in biofilm formation between Δ*gacS*

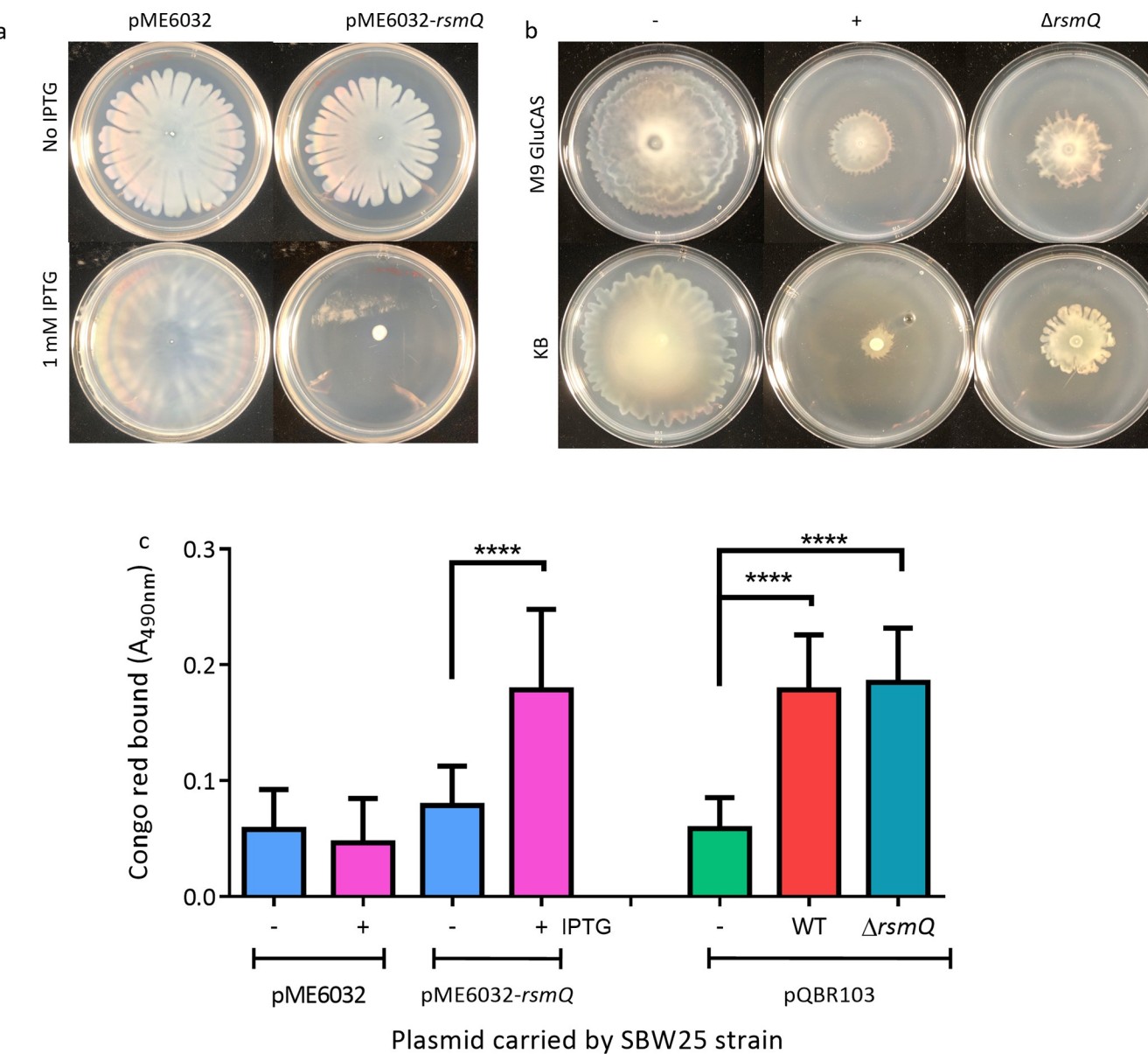

**Fig 6. Motility and biofilm formation are impacted by RsmQ.** (a) The 48-h swarming motility assays for SBW25 containing pME6032 +/- *rsmQ*. (b) The 72-h swarming motility assays for SBW25 cells either plasmid free (-) or carrying pQBR103$^{Km}$ (+) or pQBR103$^{Km}$-Δ*rsmQ* grown on 0.5% agar plates with media as indicated. Quantification of swarming of 4 biological replicates in triplicate can be found in S5 Fig. (c) Congo red absorbance (A$_{490}$) of SBW25 strains after 48 h (light blue, pink bars) or 72 h (green, red, dark blue bars). Independently performed one-way ANOVA results show statistically significant differences for both overexpression ($p < 0.0001$) and deletion ($p < 0.0001$). Statistical significance from Tukey's multiple comparisons is indicated ($p < 0.0001$, ****). Data are available in S5 Data. WT, wild-type.

and Δ*gacS* +pQBR103$^{Km}$ ±Δ*rsmQ* suggesting that the plasmid biofilm phenotype proceeds via Gac-Rsm system dysregulation, potentially mediated by other plasmid-encoded regulatory proteins.

## RsmQ influences plasmid conjugation rate in poor nutrient conditions

Next, we tested whether the phenotypes influenced by *rsmQ* had an impact on the ability of SBW25 to colonise the plant rhizosphere. The rhizosphere is a complex, heterogenous

environment, with diverse nutrient sources and physical habitats. Successful colonisation involves the coordinated deployment of several distinct ecological traits. A series of competitive wheat colonisation assays were performed between plasmid-free SBW25 cells and either SBW25 plasmid free, SBW25 +pQBR103[Km] or SBW25 +pQBR103[Km]-ΔrsmQ. After 7 days, significantly fewer colony-forming units were recovered for both plasmid carrying strains compared with the plasmid free competitor (Fig 7A). However, no significant difference was seen between strains with or without rsmQ.

Rsm proteins were first characterised by their involvement in carbon storage and metabolism (e.g., CsrA in E. coli) and the regulation of secondary metabolism [66,67]. To dissect the ecological relevance of RsmQ more closely, we examined bacterial fitness and conjugation rates in liquid media and solid agar, for strains grown in KB, M9 glucose + casamino acids (M9 GC) and M9 pyruvate (M9 Pyr). No significant differences were observed in either fitness or conjugation rate in liquid culture (S9 Fig). Conversely, we saw several key differences between strains grown on a solid surface. Plasmid carriage incurred an rsmQ-independent fitness cost on both nutrient-rich agars (KB and M9 GC) but not on nutrient-poor M9 Pyr (Figs 7B and S10) compared to a plasmid-free background. Furthermore, we saw a major increase in plasmid conjugation rate for cells grown on M9 Pyr compared to the rich nutrient sources (Fig 7C). Strikingly, conjugation rate was conditionally dependent on rsmQ, with a significantly ($p < 0.05$) lower rate seen for pQBR103[Km]–ΔrsmQ compared to pQBR103[Km] (Fig 7C) on M9 Pyr.

## Carbon source sensing by RsmQ

RsmQ exerts carbon-source dependent control of both conjugation rate (Fig 7C) and swarming (S7 Fig) in surface-grown cells. However, it remains unclear the extent to which RsmQ is involved in the sensing of specific carbon sources.

Carbon source utilisation was compared between plasmid-free SBW25 cells and cells carrying pQBR103[Km] ±rsmQ using BioLog PM1 and PM2A plates, with NADH production being used as a readout. Cells were grown on 190 different carbon sources to determine the metabolic changes that occur with plasmid carriage. After 48 h, several differences were observed, with the majority of changes relating to amino acid utilisation (S2 Table and Figs 8 and S11). While the results indicate that there is inherent variability, it is clear that cells carrying the pQBR103[Km]-ΔrsmQ plasmid have an altered carbon source metabolism than pQBR103[Km]. The pQBR103[Km] –ΔrsmQ carrying cells show a reduced ability to metabolise amino acids such as L-aspartic acid and L-anayl-glycine compared to both plasmid free and pQBR103[Km] cells. Interestingly, cells carrying pQBR103[Km] –ΔrsmQ are able to metabolise citric acid, D-alanine, and D-sorbitol at similar levels to WT cells, suggesting that RsmQ is able to repress metabolism of these carbon sources. This is in line with our proteomic data, which suggests that RsmQ represses amino acid transportation and metabolism in rich media conditions. However, in some cases when these are available as the sole carbon source, they are more easily metabolised by cells that have RsmQ present, supporting the idea that RsmQ is involved in the regulation of amino acid uptake and metabolism.

As well as this, the metabolism of other carbon sources typically found in plants, such as Hydroxyl-L-Proline [68] and m-Tartaric acid [69] were impacted by the loss of RsmQ (Fig 8A). This, along with the increase in D,L-malic acid (malate) metabolism suggest that there may be a link between the carbon sources utilised by plasmid carriers and their plant host species (Fig 8A). This is particularly interesting as malate is a major component of root exudates [70,71], suggesting that plasmid carriage may increase growth rate when in close proximity to plant roots.

a

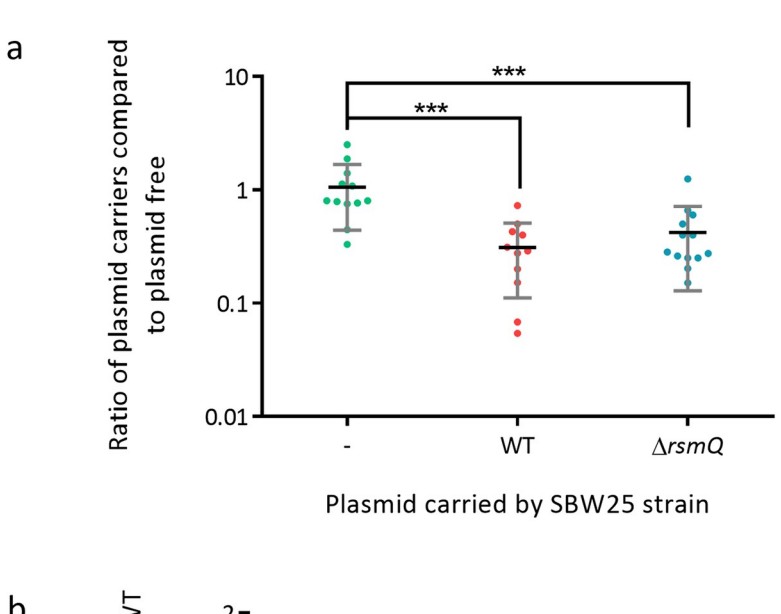

b

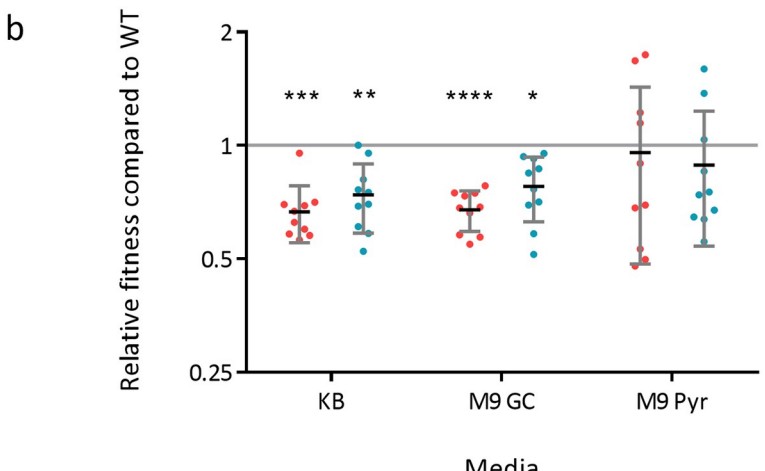

c

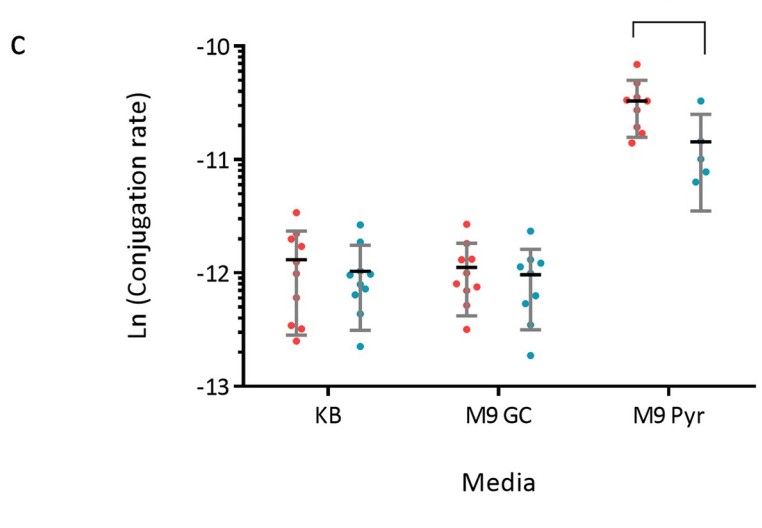

SBW25 + pQBR103$^{Km}$    SBW25 + pQBR103$^{Km}$- $\Delta rsmQ$

**Fig 7. RsmQ exerts a conditional effect on plasmid conjugation rate.** (a) Competitive rhizosphere colonisation assay comparing the fitness of plasmid-free SBW25 with SBW25 (-, green), carrying either pQBR103$^{Km}$ (WT, red) or pQBR103$^{Km}$-Δ*rsmQ* (Δ*rsmQ*, blue). Mann–Whitney U tests comparing plasmid-carrying cells to plasmid free with significance indicated ($p < 0.001$ ***). (b) Relative fitness on solid agar of plasmid-free SBW25 with SBW25 carrying either pQBR103$^{Km}$ (red) or pQBR103$^{Km}$-Δ*rsmQ* (blue) on KB, M9 GC, or M9 Pyr. Significant differences were observed between plasmid carriers and plasmid-free strains in KB and M9 GC media (indicated above as follows: $p < 0.05^*$, $p < 0.001^{**}$, $p < 0.0001^{***}$, $p < 0.00001^{****}$) when compared using a one-way ANOVA with Tukey's multiple comparisons; however, no differences were observed when comparing SBW25 +pQBR103$^{Km}$ ±Δ*rsmQ*. (c) Conjugation rate for SBW25 carrying either pQBR103$^{Km}$ (red) or pQBR103$^{Km}$-Δ*rsmQ* (blue) on KB, M9 GC, or M9 Pyr. Two-way ANOVA analysis with Sidak's multiple comparisons showed both a significant media ($p < 0.0001$) and plasmid ($p < 0.01$) effect and a significant difference between pQBR103$^{Km}$ and pQBR103$^{Km}$-Δ*rsmQ* on M9 Pyr ($p < 0.01^{**}$). All assays were performed with 10 biological and 2 technical replicates. Data containing a conjugation rate count of 0 were excluded from the dataset. Data are available in S6 Data.

To interrogate this further, we selected a subset of carbon sources shown to be metabolised differently by SBW25 containing pQBR103$^{Km}$ ±*rsmQ* in the BioLog assays and conducted growth assays in M9 minimal media with a sole carbon source (Fig 8B). Interestingly, although distinct differences were observed for several carbon sources, these did not necessarily recapitulate the BioLog results. For example, in the BioLog assays D-sorbitol, citric acid and glycerol showed an increase in metabolism for SBW25 +pQBR103$^{Km}$-Δ*rsmQ* cells compared to pQBR103$^{Km}$. However, we saw no difference in growth between plasmid carriers (Fig 8B). Although cell density is often used as a proxy for metabolism, these discrepancies may be due to underlying differences between these 2 parameters. These results confirm that pQBR103$^{Km}$ carriage directly impacts carbon metabolism; however, the impact of RsmQ appears to be directly linked to metabolism and not necessarily to growth rate.

## Discussion

By encoding homologues of bacterial regulators, plasmids can manipulate the expression of chromosomal genes and thereby alter the behaviour and phenotype of their host bacterial cells [3]. Previous studies of PCC have identified plasmid-encoded transcriptional regulators with limited and specific regulons [6]. Here, we expand the known molecular mechanisms mediating PCC to include a global post-transcriptional regulator, RsmQ. We show that RsmQ is a homologue of the widespread Csr/Rsm family of translational regulators. RsmQ alters the abundance of chemotaxis/motility and metabolism related proteins via several post transcriptional mechanisms, leading to observable growth and plasmid transmission differences in distinct carbon sources.

The soil surrounding plant roots is a complex and intensely competitive environment. Rhizosphere-dwelling bacteria respond to their surroundings at an individual level using networks of signalling proteins [72] that control bacterial behaviour, enabling effective colonisation and environmental adaptation. Simultaneously, the distribution of genes in rhizosphere metagenomes are under intense selection to best fit the prevailing environment. In recent years, substantial progress has been made towards understanding both the regulatory pathways that control bacterial rhizosphere colonisation [73], and the effect of environmental inputs on microbiome composition and species' metagenomes [74–76]. Horizontal gene transfer, for example, by conjugative plasmids, is well understood as an important driver of genetic adaptation [41,77,78]. However, the influence of plasmid-encoded regulatory genes in bacterial signalling and behaviour and the importance of this process to bacterial fitness and evolution in the rhizosphere is much less clear.

We found that numerous plasmids encode *rsm* homologues, although these were not evenly distributed, with plasmids carried by Pseudomonadaceae frequently containing *rsm* genes, while plasmids associated with other taxa had none. This distribution suggests that plasmid-

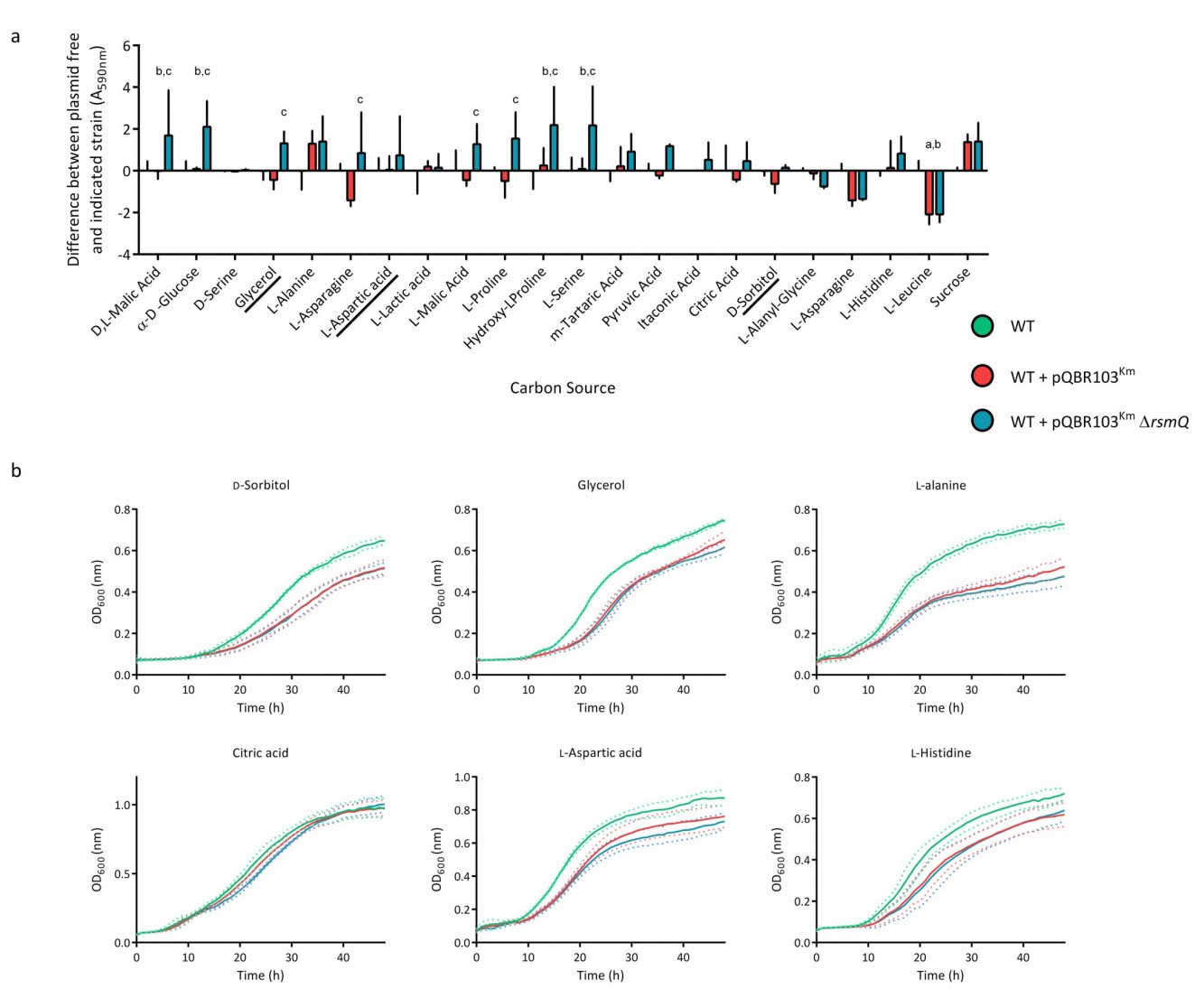

**Fig 8. RsmQ is involved in the control of carbon source metabolism.** (a) Selected results from BioLog carbon source screens showing metabolism with relevant carbon sources after 48 h of static incubation at 28°C. Data are shown for SBW25 cells carrying pQBR103Km (red) and pQBR103Km-ΔrsmQ (blue) compared to average SBW25 plasmid free (green) for each carbon source. Error bars show standard deviation. A two-way ANOVA showed a significant effect of strain and carbon source ($p < 0.0001$) as well as a significant interaction ($p < 0.01$). Multiple comparisons compared SBW25 plasmid free, SBW25 +pQBR103Km and SBW25 +pQBR103Km- ΔrsmQ within each carbon source. Significant differences are indicated above for SBW25/SBW25 +pQBR103Km (a), SBW25/SBW25 +pQBR103Km-ΔrsmQ (b), and SBW25 +pQBR103Km /SBW25 +pQBR103Km-ΔrsmQ (c) for $p < 0.05$. Data for all carbon sources can be seen in S10 Fig and S2 Table. (b) Growth curves are shown for SBW25 (green), SBW25 cells carrying pQBR103Km (red), and pQBR103Km-ΔrsmQ (blue), with the mean growth for 3 biological replicates shown as a solid line and standard deviation shown as dotted lines. Cells were grown for 48 h at 28°C without shaking in M9 minimal media with 0.4% w/v either D-sorbitol, glycerol, L-alanine, citric acid, L-aspartic acid, L-histidine as indicated. Data are available in S7 Data. WT, wild-type.

encoded Rsm proteins may fulfil important functional roles that are associated with particular groups of microbes. Plasmid carriage of *rsm* regulators appears to be a convergent trait that has emerged many times, potentially in response to similar environmental pressures affecting different plasmids. Overall, our analysis suggests that diverse plasmids have acquired *rsm* regulator genes from their bacterial hosts over time, and that these genes now appear to be evolving distinctly from their chromosomally located ancestors. Interestingly, *rsm* genes have also been

identified on bacteriophage [43], suggesting that diverse mobile genetic elements may exploit host post transcriptional regulation to ultimately serve the fitness interests of mobile genetic elements.

RsmQ-linked PCC in *P. fluorescens* SBW25 differs markedly from most recently described PCC systems, which typically alter a small number of targeted genes upon plasmid carriage [6]. In contrast, our data shows that RsmQ functions as a global, regulator of metabolism, nutrient transport, and chemotaxis pathways. RsmQ-driven regulation occurs predominantly at a post transcriptional level, affecting the abundance of hundreds of proteins and extensively subverting bacterial motility and metabolic pathways to the benefit of the plasmid. Our results implicate RsmQ and by extension other, similar plasmid-borne regulators, in considerably more extensive control of bacterial behaviour than has previously been observed.

In contrast to earlier studies [51], we saw relatively little effect of plasmid carriage or *rsmQ* on SBW25 gene transcription, likely due to the difference in test conditions used to interrogate the RsmQ function compared to earlier studies. Nonetheless, this does suggest that the nature and extent of pQBR103 control of SBW25 transcription may be highly dependent on the surrounding environment. This aligns with the phenotypic data for pQBR103$^{Km}$ carriage, where significant differences in growth rate were seen on some carbon sources but not others. Consistent with this, *rsmQ* is important for the carbon source-dependent suppression of swarming motility and the enhancement of plasmid conjugation rate, but only under certain nutrient conditions. Conjugation is a tightly controlled process within bacterial cells and environmental cues and nutrient environments can have an impact on conjugation rate [40,79]. This *rsmQ*-dependent impact of carbon source on conjugation rate on solid surfaces is fascinating, suggesting that the decision to transfer horizontally or vertically may be regulated by the plasmid in response to the surrounding environment. It is possible that plasmids up-regulate conjugation in poor media conditions (e.g., M9 Pyr), where cell replication rate is constrained, balancing reduced vertical transmission with increased horizontal transmission.

RsmQ functions in SBW25 by interacting with the host Gac-Rsm pathway in several different ways. Firstly, RsmQ binds to specific RNA target sequences at conserved short (GGA) and extended (AnGGA) binding motifs in a similar manner to chromosomally encoded Rsm proteins, especially where these are presented on the loops of ssDNA hairpins. To some extent, this RNA binding effect contributes to RsmQ-mediated PCC by directly regulating gene targets through chromosomal mRNA binding. That said, the AnGGA RsmQ-binding motif is only found upstream of around half of RsmQ-affected SBW25 genes. Thus, direct mRNA binding probably only accounts for a fraction of the observed RsmQ regulon.

At least part of the observed RsmQ regulon is likely the result of indirect proteomic adaptation in response to changes induced by RsmQ. This has been shown previously for the translational regulator Hfq in SBW25, where the direct targets of translational control do not correspond neatly to the observed proteomic regulon [9,60]. Another, potentially direct route for RsmQ function is through interaction with the host ncRNAs RsmZ and RsmY. RsmQ binds strongly to several RsmY/Z motifs as shown by SPR, suggesting that similar binding takes place in the SBW25 cell. This binding interaction could in-turn mediate pleiotropic changes in gene translation, either through RsmY/Z titration of RsmQ away from its mRNA binding targets, or alternatively via a reduction in the amount of RsmY/Z available to modulate the activity of the host Rsm proteins. Both of these alternative, and potentially antagonistic mechanisms may function simultaneously to some extent, with the abundance of RsmQ and RsmY/Z in the cell determining their relative importance to cellular Rsm function.

In addition to its interaction with host RNAs, we also saw evidence for extensive Rsm protein dimerisation, both with itself and the host Rsm proteins. Rsm protein homodimerisation is a well-characterised trait in *Pseudomonas* spp. [27,49,62]. However, to our knowledge, Rsm

proteins have not previously been shown to form heterodimers. Plasmid/host Rsm protein heterodimers may be functionally and mechanistically interchangeable, but this seems unlikely given their distinct regulons [24] and observed differences in oligonucleotide binding affinity. Alternatively, RsmQ may have an agonistic or antagonistic effect on the host Rsm proteins, and might also influence mRNA binding preference, shifting the global Rsm regulatory response towards an outcome that benefits horizontal or vertical plasmid transmission.

Carriage of pQBR103[Km] leads to an increase in biofilm formation and reduced motility, 2 Gac-system-associated traits. From the perspective of the plasmid this makes sense; tightly packed biofilms are more likely to support plasmid transmission [3]. These key phenotypes associated with plasmid carriage were also seen upon *rsmQ* overexpression. However, while *rsmQ* deletion from pQBR103[Km] partially ameliorated the plasmid-induced loss of motility, we saw no effect on biofilm, suggesting that this *rsmQ* overexpression phenotype may be non-specific. pQBR103[Km] encodes multiple putative signalling proteins in addition to *rsmQ*, including transcriptional and translational regulators and second messenger signalling enzymes. These regulators are also likely to influence SBW25 phenotypes in ways that benefit the plasmid. This may explain why *rsmQ* deletion alone only partially recovers some plasmid-associated phenotypes and has little impact on others, such as biofilm formation.

Multiple plasmid-borne regulators may function in a coordinated manner in SBW25, interacting with one another and modulating cellular responses to the environment at a global level. For example, Hfq is a global master regulator of translation [11,60] with significant regulatory overlap with the Rsm proteins [80]. While Hfq controls multiple phenotypes in SBW25 [11], the phenotypic effects of Hfq dysregulation upon *rsmQ* deletion were surprisingly modest, particularly when compared to secondary Hfq control by the RimABK system, for example [11,81]. An intriguing possibility is that the plasmid-borne Hfq homologue pQBR0137, whose mRNA abundance increases in the absence of *rsmQ*, may compensate in some way for reduced chromosomal Hfq activity. How the various plasmid-encoded regulators interact with each other and with the host regulatory network, and the impact of this on rhizosphere colonisation, is under active investigation.

In conclusion, we propose that RsmQ functions as a global PCC regulator that both directly controls host mRNA translation and interferes with the host Gac-Rsm pathway and the master regulator Hfq. RsmQ activity remodels host metabolism and suppresses motility as part of a wider PCC programme that directs *P. fluorescens* towards biofilm formation and a sessile lifestyle, where horizontal plasmid transmission is likely to be favoured. We hypothesise that core elements of this regulatory paradigm will be shared between diverse plasmid-borne Rsm proteins, with the PCC regulon tuned in each case to best support plasmid fitness in the host environment. More broadly, whereas plasmid accessory genes are often conceptualised as directly providing distinct, novel functions such as toxin efflux or enzymatic degradation of metabolic substrates, our work shows that plasmids might also find success, and indeed exert ecologically important effects, by manipulating and tuning the expression of functions encoded by genes already resident within the cell.

## Materials and methods

### Strains and growth conditions

Strains and plasmids are listed in S3 Table and primers are listed in S4 Table. Unless otherwise stated, *P. fluorescens* SBW25 were grown at 28˚C and *E. coli* strains at 37˚C in Lysogeny broth (LB) [82] solidified with 1.5% agar where appropriate. Liquid cultures were grown in 10 mL microcosms at 28˚C for *P. fluorescens* and 37˚C for *E. coli* at 250 rpm unless otherwise stated. Minimal media was made using M9 salts supplemented with 2 mM $MgSO_4$ and 0.1 mM $CaCl_2$

and each carbon source present at 0.4%. For motility assays, plates were solidified with 0.5% agar. Gentamicin (Gent) was used at 25 μg ml$^{-1}$, Streptomycin (Strep) at 250 μg ml$^{-1}$, Kanamycin (Kan) at 50 μg ml$^{-1}$, Carbenicillin (Carb) at 100 μg ml$^{-1}$, Tetracycline (Tet) at 12.5 μg ml$^{-1}$, IPTG at 1 mM, and X-gal at 40 μg ml$^{-1}$.

## Molecular biology procedures

Cloning was carried out in accordance with standard molecular biology techniques. All pTS1 plasmid inserts were synthesised and cloned into pTS1 by Twist Bioscience. The ORF of *rsmQ* was amplified by PCR with primers RsmQ_EcoRI_F and RsmQ_XhoI_R and ligated between the EcoRI and XhoI sites of pME6032. The ORF of *rsmQ* with a TEV cleavage site and a hexahistidine tag at the C-terminus was synthesised by Twist Bioscience. The ORFs of each Rsm protein were amplified by PCR using the primers indicated in S3 Table. The fragment in each case was cloned between the NdeI and XhoI sites of pET28a. Bacterial-2-hybrid plasmids were made by Gibson assembly (RsmE/I) and restriction cloning (RsmA/Q) into the BamHI and EcoRI sites of pKNT25 and pUT18C.

## Transformation of *Pseudomonas* strains

Overnight cultures of each strain were grown in LB media at 28°C, 250 rpm shaking then harvested at 6,000 ×*g* for 8 min. Cell pellets were washed twice with 0.3 M sucrose, then the pellet was resuspended in a final volume of 150 μL sucrose and placed in a 2-mm electroporation cuvette with either 2 μL of replicative plasmids or 5 μL of integrative plasmids (60 to 100 ng/μL concentration) and incubated at RT for 2 min. Cells were electroporated at 2.5 kV and recovered in 3-mL LB medium for 3 h before being plated onto LB agar containing the appropriate antibiotic. Plates were incubated for 24 to 48 h at 28°C and transformed colonies re-streaked onto fresh selective media.

## Conjugations of pQBR103$^{Km}$

Donor (*P. fluorescens* SBW25 ΩStrep$^R$-LacZ +pQBR103$^{Km}$ and *P. fluorescens* SBW25 ΩStrep$^R$-LacZ +pQBR103$^{Km}$-Δ*rsmQ*) and recipient strains (*P. fluorescens* SBW25 ΩGent$^R$ WT) were plated onto their respective selective antibiotics. Overnight cultures were set up in LB medium for each of the strains and grown overnight. Approximately 10-mL glass microcosms of Kings broth (KB) medium were inoculated with 20 μL of the donor strain and 80 μL of the recipient strain and incubated overnight without shaking. Approximately 50 μL of this overnight culture was plated onto LB medium supplemented with Gent and Kan.

## Allelic exchange

Deletion constructs were created by Twist bioscience and extracted from *E. coli* DH5-α cells. *P. fluorescens c*ells were transformed as above and incubated for 48 h until colonies appeared. Colonies were re-streaked to single colonies on fresh LB-Tet agar and incubated for 24 h. A single colony was picked and grown overnight in 50 mL of LB medium (containing kanamycin for pQBR103$^{Km}$ allelic exchange). The culture was serially diluted, and the $10^{-5}$ to $10^{-8}$ dilutions were plated onto LB agar plates containing 10% sucrose (and kanamycin for pQBR103$^{Km}$ allelic exchange). Single colonies on sucrose plates were checked for tetracycline sensitivity and confirmed as mutants by PCR.

## Swarming motility assays

Motility plates were made with 20 mL of sterile 0.5% agar in M9 GC (M9 minimal media supplemented with 0.4% glucose and 0.4% casamino acids) media unless otherwise indicated and

dried for 1 h in a laminar flow hood, rotated 180 degrees after 30 min. Approximately 3 μL of an overnight culture adjusted to an $OD_{600} = 1$ was spotted onto the centre of the plate and the lid replaced. Plates were incubated face up at room temperature for up to 72 h undisturbed and then imaged. For overexpression strains, filter sterilised 0.5 mM IPTG and tetracycline were added to the induced samples. Swarming motility was quantified using ImageJ, with the diameter of the colony at the widest part and the diameter of the plate at the widest part being measured.

## Congo red binding assays

Cultures of each strain were grown overnight in LB microcosms with selection. For each biological replicate, five 10-μl spots were placed onto 20-mL KB agar plates and incubated for 72 h. For overexpression strains, filter sterilised 0.5 mM IPTG and tetracycline were added to the induced samples. Colonies were removed from the plate and placed into 1 mL of 0.003% w/v sterile Congo red solution and incubated at 37˚C, 200 rpm shaking for 2 h. Cell material was removed by centrifugation (13,000 ×$g$ for 5 min) and absorbance was measured at 490 nm using a SPECTROstar nano plate reader (BMG). Absorbance measures were taken for 2 technical replicates.

## BioLog carbon source screening

Two colonies were picked from freshly streaked LB plates and resuspended into IF-0 inoculating fluid as per the manufacturer's instructions. PM1 and PM2A plates were inoculated with 100 μl of inoculum and incubated at 28˚C for 24 h. Plates were imaged and the absorbance at 590 nm was read on an EON microplate reader.

## Growth rate assays

Cultures of each strain were grown overnight in LB microcosms with selection. Cells were harvested at 8,000 ×$g$ and washed in M9 media without a carbon source twice. Cells were then resuspended in M9 media with each carbon source (0.4% w/v) at a starting $OD_{600}$ of 0.01 in a 96-well plate. Measurements were taken every 30 min for up to 48 h on a FLUOstar nano plate reader (BMG) with the plate being incubated at 28˚C and shaken for 2 s before each reading.

## Fitness and conjugation rate assays

Paragon wheat seeds were sterilised with 5% sodium hypochlorite and 70% ethanol then germinated on water agar in darkness for 3 days. Germinated seedlings were placed into sterile 50-ml plastic tubes containing washed, medium grain vermiculite and rooting solution (1 mM $CaCl_2 \cdot 2H_2O$, 100 μM KCl, 800 μM $MgSO_4$, 10 μM FeEDTA, 35 μM $H_3BO_3$, 9 μM $MnCl_2 \cdot 4H_2O$, 0.8 μM $ZnCl_2$, 0.5 μM $Na_2MoO_4 \cdot 2H_2O$, 0.3 μM $CuSO_4 \cdot 5H_2O$, 6 mM $KNO_3$, 18.4 mM $KH_2PO_4$, and 20 mM $Na_2HPO_4$). Tubes were transferred to a controlled environment room at 25˚C with a 16-h light cycle and grown for 7 days, then inoculated with a 1:1 mix of $1 \times 10^3$ CFU SBW25 WT ΩGent$^R$ (plasmid free, +pQBR103$^{Km}$ or +pQBR103$^{Km}$-Δ$rsmQ$) and WT Ωstrep$^R$-LacZ plasmid free. Wheat plants were grown for a further 7 days, after which the shoots were removed. A total of 20 ml of PBS was added to each tube and vortexed for 10 min at 4˚C. Five to 8 plants per condition were collected and 100 μl of the PBS suspension was diluted and plated onto LB supplemented with 100 μg/ml carbenicillin, 0.1 mM IPTG, and 50 μg/ml X-gal. Plates were incubated at 28˚C until blue and white colonies were clearly distinguishable. Colony counting was undertaken and percentage fitness calculated in each case. Experiments were repeated 3 times independently. For fitness and conjugation

experiments, 10 independent transconjugants (donor) and 10 independent colonies (recipient) were grown in 10 mL of LB overnight (+ kanamycin for plasmid containing strains). Cultures were washed twice in KB or M9 salts for KB and M9 competition experiments, respectively. The $OD_{600}$ of each culture was adjusted to 1.0, then donor (WT Strep$^R$+pQBR103$^{Km}$/WT Strep$^R$ +pQBR103$^{Km}$-$\Delta rsmQ$) and recipient (WT Gent$^R$) cells were mixed at a 1:1 ratio with 60 µl of this added to 6 mL liquid cultures of KB, M9 GC or M9 Pyr were inoculated with the initial starting mixture and incubated for 48 h (KB and M9 GC) or 96 h (M9 Pyr) at 28˚C, shaking at 180 rpm. Cells were plated on LB with selection and incubated overnight before CFU counting. Relative fitness and conjugation rate were calculated as in [51]. For solid media, 20 mL plates of KB, M9 GC, and M9 Pyr supplemented with 1.5% agar were inoculated with 10 µL of a 1:1 ratio of donor and recipient as above. Plates were incubated for 4 days (KB and M9 GC) and 8 days (M9 Pyr), before colonies were removed and resuspended in 1 mL liquid media. Final CFU measurements and calculations were performed as above.

## Bacterial 2 hybrid assays

The ORF of RsmA/E/I/Q were cloned into pKNT25 and pUT18C using either conventional restriction enzyme cloning or Gibson assembly using standard manufacturers protocols as indicated in S4 Table. Chemically competent BTH101 cells were co-transformed with both a pUT18 and a pKNT25 plasmid containing the ORF of the protein of interest using the heat shock method. Briefly, cells were incubated on ice with the plasmids for 30 min, followed by a 45-s incubation at 42˚C followed by 5 min on ice. Cells were recovered in 6 volumes of SOC media for 1 h and plated onto LB agar supplemented with Carb, Kan, and 0.5 mM IPTG. A total of 5 mL LB+Carb+Kan microcosms were grown overnight at 28˚C. Approximately 100 µL of this overnight culture was used for the β-galactosidase assay and 5 µL spots were placed onto LB+Carb+Kan+X-gal+IPTG plates and incubated overnight before being imaged.

## β-galactosidase assays

*E. coli* BTH101, 5 mL microcosms of cells carrying both pUT18C and pKNT25 plasmids were grown at 28˚C. A total of 100 µL of this was taken and incubated with 900 µL lysis buffer (60 mM $Na_2HPO_4.7H_2O$, 40 mM $NaH_2PO_4.H_2O$, 10 mM KCl, 1 mM $MgSO_4$, 7.7 mM β-mercaptoethanol, 0.001% SDS) and 20 µL chloroform at 28˚C for >10 min until cells lysed. Approximately 200 µL of 4 mg mL$^{-1}$ ONPG was added, and samples monitored until the substrate had turned yellow. To stop the reaction, 500 µL of 1M $Na_2CO_3$ was added, and the absorbance was taken at 420 and 550 nm using a FLUOstar plate reader (BMG) and $OD_{600}$ of each sample was measured using a spectrophotometer. The Miller units were calculated using the standard calculation [82].

## Protein purification

The ORF of *rsmQ* was synthesised with a C-terminal TEV cleavage site extension and a 6× histidine tag (Twist bioscience) and cloned into pET29a between the NdeI and XhoI sites. The plasmid was transformed into BL21(DE3) pLysS (Promega) by heat shock. Approximately 2.5 L of culture was inoculated at 1:50 from an overnight culture and grown until mid-log phase ($OD_{600}$ = approximately 0.6). Cultures were induced with 1 mM IPTG and grown for 16 h at 37˚C. Cells were harvested at 6,000 ×*g*, 4˚C for 15 min, and resuspended in binding buffer (20 mM Tris-HCl, 500 mM NaCl, 10 mM imidazole, 5% glycerol (pH 7.5)) containing 1 mg ml$^{-1}$ lysozyme (sigma), 1 complete protease inhibitor tablet EDTA-free and 5 µl DNaseI (Promega), lysed using a cell disruptor and the insoluble fraction removed by centrifugation (15,000 ×*g*, 25 min, 4˚C). The soluble fraction was loaded onto a HisTrap HP 5-mL column (Cyvitia) and

washed with binding buffer (20 mM Tris-HCl, 500 mM NaCl, 2.5% glycerol (pH 7.5)) with 50 mM imidazole to remove nonspecific contaminants. Proteins were eluted over a gradient of 50 to 500 mM imidazole and fractions analysed by SDS-PAGE.

Fractions containing RsmQ were dialysed overnight at 4˚C into SEC buffer (50 mM Tris-HCl, 200 mM NaCl, 2.5% Glycerol (pH 7.5)) and further purified by size exclusion chromatography using an Superdex S75 column (Cyvitia). Fractions were analysed by SDS-PAGE and pure fractions were concentrated to 3 mg ml$^{-1}$ and stored at −80˚C until needed.

For H43A and R44A, plasmids were created by site-directed mutagenesis PCR using overlapping primers (RsmQ_H43A_F, RsmQ_H43A_R, RsmQ_R44A_F, RsmQ_R44A_R), confirmed by sequencing (Source bioscience), and purified as for the WT.

## Surface plasmon resonance

ssDNA with the ReDCaT linker region at the 3′ end were synthesised (IDT) and diluted to a final concentration of 1 mM in HBSEP+ buffer (10 mM HEPES, 150 mM NaCl, 3 mM EDTA, and 0.05% v/v Tween 20 (pH 7.4)). All primer sequences can be found in S4 Table (ReDCaT). RsmQ, RsmQ H43A, and RsmQ R44A were diluted to 1,000 and 100 nM concentrations in HBSEP+ buffer. SPR measurements were recorded at 20˚C using a Biacore 8k system using a ReDCaT SA sensor chip (GE Healthcare) with 8 immobilised channels as described in (48]. RsmQ interaction was first analysed using an affinity method to examine presence and absence of binding to each of the oligos.

## RNAseq

Five independent conjugations were set up for each biological replicate as described above. A single colony from each conjugation event was picked and grown overnight in 10 mL KB medium with Gent and Kan at 28˚C, 230 rpm. The OD$_{600}$ of these cultures was measured and 60 mL KB cultures were set up at OD$_{600}$ = 0.01. Cultures were grown at 28˚C, 230 rpm until OD$_{600}$ = 1.4. Approximately 2 mL of this culture was harvested for RNA extraction and 50 mL was taken for whole proteome analysis. Pellets were collected at 8,000 ×$g$, 10 min at 4˚C and flash frozen in liquid nitrogen before storage at −80˚C.

For RNAseq, the pellets were resuspended in 150 μL 10 mM Tris-HCl (pH 8) and mixed with 700 μL of ice cold RLT+BME (RLT buffer (Qiagen) supplemented with 1% β-mercaptoethanol) and cells were lysed using a Fastprep (MP Bio) using Lysis matrix B beads (MP Bio). Lysis matrix was removed by centrifugation (13,000 ×$g$, 3 min) and the supernatant was added to 450 μL of ethanol. The supernatant was applied to an RNeasy column and RNA extraction was performed as per the manufacturer's instruction including the on-column DNA digest. After extraction, a Turbo DNase (Promega) digest was performed as per the manufacturer's instruction and total RNA yield was quantified using a Qubit RNA broad spectrum assay kit as per the manufacturer's instructions. Library construction, rRNA depletion, and paired-end Illumina sequencing (Novaseq 6,000, 2 × 150 bp configuration) were performed by Novogene. Reads provided by Novagene (as fastq.gz files) were mapped to the genome of *P. fluorescens* (NCBI accession AM181176.4) and the plasmid pQBR103 (NCBI accession AM235768.1) using the "subread-align" command of the Subread package [83]. The resulting. bam files were then sorted and indexed using the appropriate functions from the Samtools package [84]. A custom Perl script was used to make a saf file for all the gene in the genome and the plasmid. The "featureCounts" tool of the Subread package was used to count the reads mapping to every gene. The counts were read into a DGEList object of the Bioconductor package edgeR and a quasi-likelihood negative binomial generalized log-linear model was fitted to the data using the "glmQLFit" function of edgeR [85]. Genewise statistical tests were

conducted using the "glmQLFTest" function of edgeR. Finally, the "topTags." Processed data is deposited in ArrayExpress (E-MTAB-11868).

## qRT-PCR

Cultures of each strain were grown overnight in 6 mL KB microcosms with selection. A total of 50 mL KB flasks without selection were inoculated 1:50 with 2 mL samples being taken every hour from 2 to 8 h with pellets being flash frozen in liquid nitrogen. Optical density at 600 nm was measured at each time point to determine growth phase. Time points were selected at mid-log, late-log, and early stationary phase. RNA was extracted as above and quantified using a Qubit as above.

qRT-PCR was performed using the Luna universal one-step qRT-PCR kit (NEB) as per the manufacturer's instructions. Briefly, in a 20 μL reaction, 200 ng of RNA, 0.4 μM forward primer (S4 Table), 0.4 μM reverse primer (S4 Table), 1× reaction buffer, and 1× enzyme mix was added. qRT-PCR was performed with a reverse transcriptase step of 10 min at 55˚C and a 1 min 95˚C denaturing step, followed by 45 cycles of 95˚C for 10 s and 60˚C for 30 s. Melt curves were performed from 65 to 95˚C in 0.5˚C increments. Ct values were compared to the reference sample and analysed by one-way or two-way ANOVAs where appropriate.

## Whole proteome analysis

Cells were grown as above and stored at −80˚C until the proteome was extracted. Three samples were thawed on ice and resuspended in ice cold 500 μL lysis buffer (20 mM Tris, 0.1 M NaCl, pH 8 + 1 complete protease inhibitor tablet). Cells were lysed by sonication at 12 mA (1 s on, 3 s off for 20 cycles). Insoluble material was removed by centrifugation at 4˚C, 4,000 ×*g*, 10 min. The supernatant was taken and the proteome precipitated for 10 min with the addition of 8 volumes of acetone at RT. The proteome was pelleted by centrifugation at 7,000 ×*g* for 10 min and washed once more with acetone. Protein pellets were resuspended in 400 μl of 2.5% sodium deoxycholate (SDC; Merck) in 0.2 M EPPS-buffer (Merck), pH 8.5, and vortexed under heating for a total of 3 cycles. Protein concentration was estimated using a BCA assay and approximately 200 μg of protein per sample was reduced, alkylated, and digested with trypsin in the SDC buffer according to standard procedures. After the digest, the SDC was precipitated by adjusting to 0.2% TFA, and the clear supernatant subjected to C18 SPE (OMIX tips; Agilent). Peptide concentration was further estimated by running an aliquot of the digests on LCMS (see below). TMT labelling was performed using a TMTpro 16plex kit (Lot WB314804, Thermo Fisher Scientific) according to the manufacturer's instructions with slight modifications; approximately 100 μg of the dried peptides were dissolved in 90 μl of 0.2 M EPPS buffer (MERCK)/10% acetonitrile, and 250 μg TMT reagent dissolved in 22 μl of acetonitrile was added. Samples were assigned to the TMT channels.

After labelling, aliquots of 1.5 μl from each sample were combined in 500 μl 0.2% TFA, desalted, and analysed on the mass spectrometer (see below) to check labelling efficiency and estimate total sample abundances. The main sample aliquots were then combined correspondingly to roughly level abundances and desalted using a C18 Sep-Pak cartridge (200 mg, Waters). The eluted peptides were dissolved in 300 μl 0.1% TFA and fractionated with the Pierce High pH Reversed-Phase Peptide Fractionation Kit (Thermo Fisher Scientific) according to the manufacturer's instructions. Fractions for the mass spectrometry analysis were eluted sequentially with the following concentrations of acetonitrile: 7.5%, 10%, 12.5%, 15%, 17.5%, 20%, 30%, 40%, 50% and dried down and resuspended in 0.1% TFA, 3% acetonitrile.

Aliquots were analysed by nanoLC-MS/MS on an Orbitrap Eclipse Tribrid mass spectrometer coupled to an UltiMate 3000 RSLCnano LC system (Thermo Fisher Scientific). The

samples were loaded onto a trap cartridge (PepMap 100, C18, 5 μm, 0.3 × 5 mm, Thermo) with 0.1% TFA at 15 μl min$^{-1}$ for 3 min. The trap column was then switched in-line with the analytical column (nanoEase M/Z column, HSS C18 T3, 1.8 μm, 100 Å, 250 mm × 0.75 μm, Waters) for separation using the following gradient of solvents A (water, 0.1% formic acid) and B (80% acetonitrile, 0.1% formic acid) at a flow rate of 0.2 μl min$^{-1}$: 0 to 3 min 3% B (parallel to trapping); 3 to 10 min linear increase B to 8%; 10 to 90 min increase B to 37%; 90 to 105 min linear increase B to 50%; followed by a ramp to 99% B and re-equilibration to 3% B. Data were acquired with the following parameters in positive ion mode: MS1/OT: resolution 120K, profile mode, mass range m/z 400 to 1,800, AGC target 100%, max inject time 50 ms; MS2/IT: data dependent analysis with the following parameters: top10 in IT Rapid mode, centroid mode, quadrupole isolation window 0.7 Da, charge states 2–5, threshold 1.9e4, CE = 30, AGC target 1e4, max. inject time 50 ms, dynamic exclusion 1 count for 15-s mass tolerance of 7 ppm; MS3 synchronous precursor selection (SPS): 10 SPS precursors, isolation window 0.7 Da, HCD fragmentation with CE = 50, Orbitrap Turbo TMT and TMTpro resolution 30k, AGC target 1e5, max inject time 105 ms, real time search (RTS): protein database *Pseudomonas fluorescens* SBW25 (uniprot.org, 02/2016, 6,388 entries), enzyme trypsin, 1 missed cleavage, oxidation (M) as variable, carbamidomethyl (C), and TMTpro as fixed modifications, Xcorr = 1, dCn = 0.05.

The acquired raw data were processed and quantified in Proteome Discoverer 2.4.1.15 (Thermo Fisher Scientific) using the incorporated search engine Sequest HT and the Mascot search engine (Matrix Science; Mascot version 2.8.0). The processing workflow included recalibration of MS1 spectra (RC), reporter ion quantification by most confident centroid (20 ppm), fasta databases *P. fluorescens* SBW25 (as for RTS) and common contaminants, precursor/fragment tolerance 6 ppm/0.6 Da, enzyme trypsin with 1 missed cleavage, variable modification was oxidation (M), fixed were carbamidomethyl (C) and TMTpro 16plex. The consensus workflow included the following parameters: unique peptides (protein groups), intensity-based abundance, TMT channel correction values applied (WB314804), co-isolation/ SPS matches thresholds 50%/70%, normalisation on total peptide abundances, protein abundance-based ratio calculation, missing values imputation by low abundance resampling, 2 or 3 replicates per sample (non-nested), hypothesis testing by *t* test (background based), adjusted *p*-value calculation by BH method. Experimental data is deposited in ProteomeXchange (PXD033640).

### Bioinformatics and sequence analysis

The COMPASS database [42] was downloaded from https://github.com/itsmeludo/ COMPASS and annotated using PROKKA 1.14.6 [86] using the default settings to identify plasmid-borne *rsmQ/csrA* genes. PROKKA uses BLAST+ matches with the curated Uni-ProtKB (SwissProt) databases to annotate proteins, followed by hidden Markov model (HMM)-based searches. To compare chromosomal and plasmid *csrA/rsmA* genes, chromosomal sequences associated with the COMPASS plasmids were identified and downloaded using NCBI elink and efetch tools and were reannotated using PROKKA to ensure comparability between plasmid and chromosomal sequences. *csrA/rsmA* sequences which were identical at the nucleic acid level were removed before conducting analyses (there were no identical sequences between chromosomes and plasmids). Start codons were unified across CsrA/ RsmA homologues by manual examination and sequence editing. Sequences were aligned by codon alignment in PRANK v.170427 [87] using the default settings. Initial alignments had a high proportion (>70%) of gaps owing to sequence divergence towards the 3′ end of the gene, which interfered with phylogenetic analysis. Trees were therefore built using information from

conserved sites, by removing columns from the alignment that consisted of majority (>60%) gaps. Duplicate sequences were removed. Trees were estimated using RAxML 8.2.12 [88] with the settings -f a -m GTRCAT -p 12345 -x 12345 -# 100. Qualitatively similar outcomes were obtained if gap columns and/or duplicate sequences were retained. Jukes–Cantor distance matrices were extracted from the alignments for analysis using the EMBOSS 6.6.0.0 and distances were compared using Bonferroni-corrected Wilcoxon tests. The script CSRA_TAR-GET.pl [89] was adapted to predict binding sites for *csrA/rsmA* in 5′ untranslated regions of the PROKKA-predicted plasmid genes, and differences in distributions of number of sites/ CDS were compared between *csrA/rsmA*-encoding and non-encoding plasmids using two-sample Kolmogorov–Smirnov tests. Analyses were performed in R (4.1.2, R Core Team, Vienna, Austria), bash, and Python 3.6 within the RStudio IDE (RStudio Team, Boston, United States of America) with the assistance of tidyverse [90] and ggtree [91] packages. Example scripts and analyses can be found on github (jpjh/PLASMAN_RsmQ).

## Supporting information

**S1 Fig.** (a) Across Families, CsrA/RsmA-encoding plasmids are relatively large. Size density plots for all Families with >20 plasmids and ≥1 plasmid-encoded CsrA/RsmA homologue. Each row describes a different Family. Semi-transparent triangles indicate the size of CsrA/RsmA-encoding plasmids. On the left, the proportion of total plasmids encoding CsrA/RsmA homologues for that Family. (b) Comparison of putative CsrA/RsmA-regulated gene frequencies between CsrA/RsmA-encoding and non-encoding Pseudomonadaceae plasmids. Plots show overlayed histograms and density plots. Left-hand plot shows absolute numbers of putative CsrA/RsmA-regulated genes, whereas right-hand plot shows as a proportion of total CDS on that plasmid. Distributions were significantly different between plasmid types in both panels (Kolmogorov–Smirnov test, $p < 0.001$ for absolute counts, $p = 0.012$ for proportions). (c) Taxonomic distribution of plasmid borne *csrA/rsmA* homologues identified in COMPASS as in Fig 1. The paler part of each stacked bar indicates genes that were identical at the nucleotide level to other identified homologues. All data and analyses are available on github (PLASMAN_RsmQ) with data taken from the COMPASS database.
(TIF)

**S2 Fig.** (a) Percentage $R_{max}$ values for RsmQ binding to ssDNAs containing the indicated binding site sequence in either a linear format (pink) or at the top of a hairpin loop (purple). (b) Percentage $R_{max}$ values for RsmE binding to a selection of ssDNAs compared to RsmQ for selected key oligos. All assays were performed in duplicate and data for 100 mM shown. (c) Percentage $R_{max}$ values for RsmE binding to all synthesised oligos with non-hairpin oligos indicated (*). Error bars on all graphs show standard deviation of 2 independent replicates at a concentration of 100 nM. Data are available in S8 Data.
(TIF)

**S3 Fig. Volcano plots highlighting RNAseq data comparing plasmid carriers and plasmid-free cells with significantly changed mRNA abundances highlighted in blue and red for (a) SBW25 + pQBR103$^{Km}$/SBW25, (b) SBW25 +pQBR103$^{Km}$-Δ*rsmQ*/SBW25, (c) SBW25 +-pQBR103$^{Km}$-Δ*rsmQ*/ SBW25 +pQBR103$^{Km}$, (d) Overlay of SBW25 +pQBR103$^{Km}$/SBW25 and SBW25 +pQBR103$^{Km}$- Δ*rsmQ*/SBW25 with $p < 0.05$ highlighted by a dashed line.** Data are available in S9 Data and publicly on ArrayExpress (E-MTAB-11868).
(TIF)

**S4 Fig. Predicted secondary structure of key oligos from Fig 4 using mFold.** Secondary structure predictions of key oligos showing the positioning of the AnGGA binding site within

the hairpin loops. Images generated using IDT oligo analyser.
(TIF)

**S5 Fig. qRT-PCR results for (a)** *rsmA,* **(b)** *rsmQ,* **(c)** *rsmE,* **(d)** *rsmI* **at 3 time points with data showing the difference between the control gene (RpoS) and the target gene (RsmA/ E/I/Q) for SBW25 plasmid free (-, green), SBW25 +pQBR103**[Km] **(WT, red), and SBW25 +- pQBR103**[Km]**-Δ***rsmQ* **(Δ***rsmQ,* **blue) with means shown as dots, and error bars showing standard deviation.** No results were obtained for RsmQ in either plasmid free or Δ*rsmQ* strains. Data are available in S10 Data.
(TIF)

**S6 Fig. Quantification of swarming motility of plasmid carrying cells over 72 h.** Swarming motility after 24 (a), 48 (b), and 72 (c) h for SBW25 cells carrying either pQBR103[Km] (WT, red) or pQBR103[Km]-Δ*rsmQ* (Δ*rsmQ*, blue) grown on 0.5% KB or M9 media with glucose casaminoacids (M9). Data points are shown for 4 biological replicates performed in triplicate, with mean values indicated with a bar. Error bars represent mean ± standard deviation. Data at each time point was compared using a two-way ANOVA, with an overall plasmid effect being seen at 24 h ($p < 0.05$), and Sidak's multiple comparisons showing a significant difference between WT and Δ*rsmQ* ($p < 0.05$, *). A significant media effect was seen at every time point ($p < 0.001$). Data are available in S11 Data.
(TIF)

**S7 Fig.** (a) Swarming motility after 72 h for SBW25 cells either plasmid free (-) or carrying pQBR103[Km] (+) or pQBR103[Km]-Δ*rsmQ* grown on 0.5% M9 media with the carbon source indicated.
(TIF)

**S8 Fig. Congo red absorbance (A**$_{490}$**) of SBW25 lacking either a single chromosomal** *rsm* **or** *gacS.* Each strain is shown as either plasmid free (green), + pQBR103[Km] (red), and + pQBR103[Km]- Δ*rsmQ* (blue) after 72 h of incubation at 28˚C on KB agar. Three independent biological replicates and 10 technical replicates were tested for each strain with error bars representing standard deviation. A two-way ANOVA results show significant effects of both strain and plasmid carriage ($p < 0.0001$,****). Sidak's multiple comparisons show significant differences between plasmid-free and plasmid-carrying strains for every strain apart from SBW25 Δ*gacS*. No significant differences were seen between plasmid-carrying strains irrespective of presence or absence of *rsmQ*. Data are available in S12 Data.
(TIF)

**S9 Fig. β-galactosidase assay (as shown in Fig 5) with all controls shown.** Data are available in S4 Data.
(TIF)

**S10 Fig. RsmQ fitness and conjugation rates in a variety of liquid media.** (a) Relative fitness in liquid media of plasmid-free SBW25 with SBW25 carrying either pQBR103[Km] (red) or pQBR103[Km]-Δ*rsmQ* (blue) on KB, M9 GC, or M9 Pyr. (b) Conjugation rate for SBW25 carrying either pQBR103[Km] (red) or pQBR103[Km]-Δ*rsmQ* (blue) on KB, M9 GC, or M9 Pyr. Data points are shown for 10 independent biological replicates with the mean (black) and standard deviation (grey) indicated by bars. Data are available in S13 Data.
(TIF)

**S11 Fig. Carbon source utilisation results for SBW25 plasmid free (green), SBW25 +- pQBR103**[Km]**, and SBW25 +pQBR103** [Km]**–Δ***rsmQ* **for 3 independent biological replicates were tested for each strain.** Absolute values are shown here for each carbon source as

individual data points with selected results shown in Fig 8A. All data are available in S2 Table.
(TIF)

**S1 Table. Up- and down-regulated proteins in for pQBR103$^{Km}$-Δ*rsmQ*/pQBR103$^{Km}$.**
(XLSX)

**S2 Table. BioLog results for SBW25 plasmid free, SBW25 +pQBR103$^{Km}$ and pQBR103$^{Km}$-Δ*rsmQ* after 24 h of growth.** Absorbance measured at 590 nm.
(XLSX)

**S3 Table. Table of plasmids and strains.**
(DOCX)

**S4 Table. Table of primers.**
(XLSX)

**S1 Data. Percentage R$_{max}$ values for RsmQ binding to predicted binding sites.** Contains underlying data for Fig 2.
(XLSX)

**S2 Data. Integrated RNAseq and proteomic data comparing pQBR103$^{Km}$±*rsmQ*.** Contains underlying data for Fig 3.
(XLSX)

**S3 Data. Percentage R$_{max}$ values for RsmQ binding to the upstream regions of predicted mRNA targets.** Contains underlying data for Fig 4.
(XLSX)

**S4 Data.** (a) Percentage R$_{max}$ values for RsmQ binding to portions of RsmY and Z. (b) Quantification of bacterial-2-hybrid β-galactosidase assays for dimerization of Rsm proteins. Contains underlying data for Figs 5 and S9.
(XLSX)

**S5 Data. Congo red absorbance (A$_{490}$) of SBW25 plasmid free and SBW25 carrying pQBR103$^{Km}$±*rsmQ*.** Contains underlying data for Fig 6.
(XLSX)

**S6 Data. RsmQ fitness and conjugation rates in planta and on solid media.** Contains underlying data for Fig 7.
(XLSX)

**S7 Data. Growth data for SBW25 plasmid free and SBW25 carrying pQBR103$^{Km}$±*rsmQ*.** Contains underlying data for Fig 8.
(XLSX)

**S8 Data. Percentage R$_{max}$ values for RsmQ and RsmE binding to predicted binding sites.** Contains underlying data for S2 Fig.
(XLSX)

**S9 Data. RNAseq data for SBW25 plasmid free and SBW25 carrying pQBR103$^{Km}$±*rsmQ*.** Contains underlying data for S3 Fig.
(XLSX)

**S10 Data. qRT-PCR results for the expression of all *rsm* genes.** Contains underlying data for S5 Fig.
(XLSX)

**S11 Data. Quantification of swarming motility for supporting data for SBW25 plasmid free and SBW25 carrying pQBR103^Km^±*rsmQ*.** Contains underlying data for S6 Fig.
(XLSX)

**S12 Data. Congo red absorbance (A_{490}) of SBW25 lacking either a single chromosomal *rsm* or *gacS*.** Contains underlying data for S8 Fig.
(XLSX)

**S13 Data. RsmQ fitness and conjugation rates in liquid media.** Contains underlying data for S10 Fig.
(XLSX)

## Acknowledgments

The authors would like to thank Clare Stevenson and Julia Mundy for their help and advice with initial SPR experiments and analysis as well as Kate Baker and Charlotte Chong for their ideas and feedback on bioinformatic analysis.

## Author Contributions

**Conceptualization:** Catriona M. A. Thompson, James P. J. Hall, Richard H. Little, Ainelen Piazza, Ellie Harrison, Robert W. Jackson, Michael A. Brockhurst, Jacob G. Malone.

**Data curation:** Catriona M. A. Thompson, James P. J. Hall.

**Formal analysis:** Catriona M. A. Thompson, James P. J. Hall, Govind Chandra, Gerhard Saalbach, Michael A. Brockhurst.

**Funding acquisition:** James P. J. Hall, Ellie Harrison, Robert W. Jackson, Michael A. Brockhurst, Jacob G. Malone.

**Investigation:** Catriona M. A. Thompson, James P. J. Hall, Carlo Martins, Supakan Panturat, Susannah M. Bird, Samuel Ford, Ainelen Piazza, Michael A. Brockhurst.

**Methodology:** Catriona M. A. Thompson, James P. J. Hall, Carlo Martins, Gerhard Saalbach.

**Project administration:** Catriona M. A. Thompson, James P. J. Hall, Susannah M. Bird, Michael A. Brockhurst.

**Resources:** Catriona M. A. Thompson, Susannah M. Bird, Michael A. Brockhurst.

**Supervision:** Ellie Harrison, Michael A. Brockhurst, Jacob G. Malone.

**Validation:** James P. J. Hall.

**Visualization:** Catriona M. A. Thompson.

**Writing – original draft:** Catriona M. A. Thompson, James P. J. Hall, Michael A. Brockhurst, Jacob G. Malone.

**Writing – review & editing:** Catriona M. A. Thompson, James P. J. Hall, Govind Chandra, Gerhard Saalbach, Richard H. Little, Ainelen Piazza, Ellie Harrison, Robert W. Jackson, Michael A. Brockhurst, Jacob G. Malone.

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
