## [Editor Report · Decision Letter 0]

1 Aug 2022

Dear Dr. Malone, 

Thank you for submitting your manuscript entitled "Plasmid manipulation of bacterial behaviour through translational regulatory crosstalk" for consideration as a Research Article by PLOS Biology.

Your manuscript has now been evaluated by the PLOS Biology editorial staff, as well as by an academic editor with relevant expertise, and I am writing to let you know that we would like to send your submission out for external peer review.

Once your full submission is complete, your paper will undergo a series of checks in preparation for peer review. After your manuscript has passed the checks it will be sent out for review. To provide the metadata for your submission, please Login to Editorial Manager (https://www.editorialmanager.com/pbiology) within two working days, i.e. by Aug 03 2022 11:59PM.

Kind regards,

Paula

---

Senior Editor

PLOS Biology

---

## [Decision Letter · Decision Letter 1]

6 Oct 2022

Dear Dr. Malone,

Thank you for your patience while your manuscript "Plasmid manipulation of bacterial behaviour through translational regulatory crosstalk" was peer-reviewed at PLOS Biology. It has now been evaluated by the PLOS Biology editors, an Academic Editor with relevant expertise, and by several independent reviewers. 

In light of the reviews, which you will find at the end of this email, we would like to invite you to revise the work to thoroughly address the reviewers' reports.

As you will see below, the reviewers find your work interesting but they all rise issues that will need to be solved before publication at PLOS Biology. You will need to address all the reviewers concerns and this will imply performing additional experiments and adding clarifications, discussion and better analyses and statistics.

Given the extent of revision needed, we cannot make a decision about publication until we have seen the revised manuscript and your response to the reviewers' comments. Your revised manuscript is likely to be sent for further evaluation by all or a subset of the reviewers.

**IMPORTANT - SUBMITTING YOUR REVISION**

*Re-submission Checklist*

*Published Peer Review*

*PLOS Data Policy*

*Blot and Gel Data Policy*

Sincerely,

Paula

---

Senior Editor

PLOS Biology

REVIEWS:

Reviewer #1: RNA regulation.

Reviewer #2: Pseudomonas, Rsm proteins, bacterial signalling pathways.

Reviewer #3: Horizontal gene transfer, plasmids, resistance.

Reviewer #1: In this study, Thompson et al investigated the regulatory role of the RsmQ protein in Pseudomonas fluorescens. RsmQ belongs to the family of CsrA/RsmA-like RNA-binding proteins that modulate gene expression by inhibiting or activating the translation of mRNAs. CsrA-like proteins are highly conserved and have been studied in multiple systems. The activity of CsrA-like proteins is counteracted by several non-coding RNAs, e.g. RsmY and RsmZ, which contain multiple CsrA binding sites and thus serve as sponges to sequester CsrA in the cell. RsmQ differs from previously studied CsrA homologues as it is located on a large conjugative plasmid (pQBR103). 

 To study the role of RsmQ in P. fluorescens, the authors tested the binding affinities of RsmQ in vitro and determined the transcriptomic and proteomic changes associated with the deletion of the rsmQ gene in vivo. Further, they discovered that RsmQ can form heterodimers with other CsrA-like proteins in the cell and tested if and how RsmQ affected the physiology of P. fluorescens.

Unfortunately, the presented work comes with four major problems, which reduced my enthusiasm for this study. First, the large discrepancies between the transcriptome and the proteomic experiments have not been addressed. Second, while the in vitro experiments using ssDNA can be a useful first step to determine the binding preferences of RsmQ, in vivo co-immunoprecipitation experiments (e.g. CLIP-seq) would be more informative to address this question. Third, the observation that RsmQ forms heterodimers with other CsrA-like proteins is potentially interesting, however, remains a "loose end" as no follow-up experiments have been performed. Finally, the experiments aiming to understand the physiological changes associated with RsmQ are inconsistence and therefore difficult to interpret. Taken together, I do not consider this manuscript a strong candidate for PLOS Biology. Major and minor comments are listed below. 

Major comments:

- Introduction, 3rd paragraph, lines 54-55: I don't think this statement is correct. The plasmid-located RNA-binding proteins FopA and FinO have both been reported to interact with transcripts from the main chromosome (see Gerovac et al, RNA, 2020 and El Mouali et al, NAR 2021).

- Fig. 2: As pointed out above, I do not fully understand the motivation for these experiments. In vivo experiments such as RIP-Seq and CLIP-Seq would be more suitable to identify the interaction partners and binding preferences of RsmQ. Also, why did the authors decide to use ssDNA in these experiments rather than RNA? 

- Fig. 3 and lines 202-218: the large discrepancies between the transcriptomic and proteomic experiments are difficult to follow. In bacteria, translation repression is typically associated with transcript decay as the ribosomes fail to protect the mRNA from ribonucleolytic cleavage. Thus, one would expect that changes in the proteome should also be detectable at the transcript level (at least for the majority of transcripts).

- Lines 242-250: this paragraph is highly speculative and does not add relevant information to the paper. 

- Fig. 5: the observation that RsmQ can form heterodimers with other CsrA-like proteins is potentially interesting, however, these experiments would be more informative if they were carried out in P. fluorescens rather than using plasmid-borne overexpression in Escherichia coli. In addition, it is not clear if and how these heterodimers would affect the RNA-ligand profiles of these proteins.

- Figs. 8b and c: I do not think these results are informative given the large variation among the results. 

Minor comments:

- Lines 108-109: given that the plasmid encodes an CrsA-like protein, i.e. RsmQ, it would be interesting to know if the plasmid also encodes homologues of the RsmY and RsmZ sRNAs.

- Fig. 6b: I think it would be useful to quantify the degree of swarming observed in these assays.

- Fig. 7c: please clarify why conjugation rates are significantly higher in M9Pyr when compared to KB and M9Glc.

- Please add error bars to Fig. 8a.

Reviewer #2: The data presented in this manuscript follow up on work by the Brokhurst's lab on plasmids transmission and fitness costs and further elaborate on work by Sobrero and Valverde describing comparative genomics with RNA-binding proteins of the CsrA family in Pseudomonadaceae. In brief the authors comprehensively reviewed the presence of rsm genes on plasmids. The Rsm pathway involves translational repressors of the Rsm family of which RsmA is the best described. Here RsmQ encoded on pQBR103 plasmid is studied in more details and its impact on the expression and production of proteins in the Pseudomonas fluorescens host comprehensively reviewed as well as the phenotypic impact it has on the lifestyle of the bacterium.

Overall, the study is very solid and comfort the concept that regulatory elements carried on plasmids influence host gene expression. Here the authors make the case that with RsmQ that is mostly the proteome which is influenced and that is per se novel. The story then goes in trying to decipher how RsmQ could influence translation and proposes two explanations. One is the direct binding of RsmQ on RNA targets, including regulatory gene targets which in turn will affect broadly and indirectly gene expression. The other explanation is that RsmQ may sequester sRNAs which otherwise would prevent endogenous Rsm to exert their regulatory control. This concept provides again some level of novelty. It is even further suggested, and this is supported by bacterial two hybrid experiments, that the regulatory modulation by RsmQ could be exerted by the formation of heterodimers with endogenous Rsm.

The overall analysis ends to propose that RsmQ would mainly influence chemotactic genes and metabolism and as such has an impact which would largely depend on environmental conditions.

Despite several excellent points of conceptual novelty, the broad analysis led to several speculative statements and in the end kept a good number of speculations open, but which may need further insights. In any cases it is not clear what would be the specific contribution of RsmQ to the bacteria lifestyle switch which would not possibly be controlled by the already complex endogenous Rsm/Gac network.

Specific comments are as follows:

- Considering RsmQ has a similar binding site as compared to other RsmA-like, and that eventually it has common targets with endogenous Rsm, could the authors assess what would be the "preferred" repressor for some of the common targets. In other words what is the affinity of the different regulators for their RNA targets. This is an important question since it may address whether RsmQ interferes with host gene expression by competing with residents Rsm. The same question holds true for the concept where RsmQ binds RsmYZ. Is the affinity different as compared to the endogenous RsmA-like?

- This is minor but do the authors have any clues on what the source for pQBR103 is in the environment. It seems from their genomic analysis that RsmQ-encoding plasmids are mostly found in Legionellaceae and Pseudomonaceae. Is there any physiological explanation for this bias?

- In a growth experiment does RsmQ and the other endogenous Rsm expressed simultaneously? Can they be expressed at distinct stages which would contribute to provide them with alternative and not competitive control on target RNAs.

- It is noted that RsmQ will trigger biofilm formation and thus increases conjugative opportunities. Yet the endogenous Rsm would rather contribute to prevent biofilm formation. It is hard to reconcile this antagonism when comparing regulators which essentially have the same targets. For example, if RsmQ sequesters RsmYZ then there will be more freed RsmA and thus more repression in biofilm. I may have missed some interpretation there, but it will be worth discussing this point in more clear details.

- The inconsistencies regarding growth in different biological replicates explained by RsmQ involved in a bistable metabolic system is appealing but is rather preliminary and would need further support (Lines 377-382 and lines 502-508).

- It seems that the presence/absence of the plasmids compares quite differently when looking at plasmids with or without a functional RsmQ (Lines 312-316). That would mean that there are additional regulatory elements on the plasmid that may influence host response. This is not really discussed and deserved to be.

Reviewer #3: In this paper, Thompson et al. investigated the role of the plasmid borne RsmQ regulator in P. fluorescens. First, they evaluate how these regulators are distributed across the bacterial taxonomy. Then they evaluate how rsmQ post-transcriptional regulation modulates the abundance of nutrient acquisition, chemotaxis and metabolism proteins. Finally, the authors emphasize how these elements interact with other bacterial proteins and unveil their impact on bacterial mode of growth.

The findings exposed in this manuscript shed some light on the role of plasmids acting as regulators of bacterial physiology by adding a well-characterized new example. I particularly enjoyed the insight that plasmid and chromosomal Rsm proteins can form heterodimers and how this further compounds Rsm-mediated translational regulation.

The paper is clearly written and might greatly interest the PLOS Biology audience. However, some issues need to be solved before publication (see below for a list of specific comments). In particular, several experiments on the paper lack appropriate statistical analyses, and the interpretation of some results need to be carefully nuanced.

Specific Comments: 

- L60: 'Rsms'. Plural protein names may be confusing. Please consider rephrasing. A suggestion could be 'rsm genes encode small proteins…'

- L113. Please clarify which criteria was followed to identify rsmQ homologs. Percentage of homology and identity?

- Fig 1. It seems that in fig 1a absolute numbers are presented. It would be more informative to include the percentage of plasmids, as the total number of rsmA homologs will likely depend on the number of plasmids per genus in the database.

- Fig 1C. The figure legend is confusing. It should state somewhere that all these belong to the Pseudomonas genus. Also, I wonder if the authors found rsmA homologs with mutations and/or deletions that could potentially lead to a loss-of-function phenotype. In that case, could be a hint pointing to compensatory evolution.

- L117 states that 'over 50% of Legionellaceae (18/30) plasmids contained rsm homologues', but in Fig1a, there seem to be more than 20. Also, the text states 41 plasmids for Pseudomonas, but the figure seems to indicate way more. Please clarify.

-Fig S1. Is there a reason why there is no distribution for Rhodanobactereaceae and Thermomicrobiacea? I guess it is because there is only one plasmid per family in the database. However, from L1127 I understood that data represented corresponds to families 'with >20 plasmids and ≥1 plasmid-encoded CsrA/RsmA homologue.' Please clarify.

- L179: For clarity, please explain here why those mutants are expected to affect the conserved VHRD/E motif. Are they directly altering the motif sequence?

- L181: Statistical analyses should be performed to sustain this claim. Also, there seem to be tiny error bars associated with the Rmax measures. What do they represent? Please specify in the figure legend.

- Table S1: There are no statistical analyses of the RNA-seq and proteomic data. Without statistics is difficult to judge the significance of the results. Additionally, gene names should be standardized between the main text and table S1 (e.g., PFLU0610 vs. PFLU_0610)

- Figure 4. At 100 nM, the %Rmax of Hairpin ACGGA and GGA seem to slightly differ from those in fig1. Is this expected? Five of the ~25 oligos tested reach the 50% threshold, but most only at really high RsmQ concentrations (i.e., above 250-500 nM). Can the authors elaborate on why they believe this is physiologically relevant?

- Nanomolar is typically abbreviated as nM, not 'nm'.

- L263-266: The secondary structure predictions are not shown. I think it could be helpful to the reader to show them as supplementary figure.

- L272: It would have been nice to include a chromosomal Rsm protein in Figure 2 assays as a control. I do not think it is imperative for acceptance as it has been demonstrated elsewhere. However, it would be very appropriate to control that the binding kinetics are similar using the same experimental procedure. It could also help to put in perspective the protein concentrations used in these experiments.

- L275: A reference should be added here, and the meaning of ncRNA should be stated the first time the term is mentioned.

- Fig 5a: I think this figure would benefit from showing whether each of the ssDNA oligos contains (or not) the AnGGA or GGA motif. Again, error bars are shown but not described, and there is no statistical analysis of the results.

- L293: a sentence briefly explaining the BATCH system may help the reader understand this experiment.

- L321: It is really hard to see how the loss of rsmQ restores swarming motility in this experiment. I barely see any difference between rsmQ+ and rsmQ- plates. Similarly, in figure 6B, it is clear that the presence of pQBR103 affects swarming motility, but the deletion of rsmQ barely restores swarming. This indicates that other plasmid traits may be responsible for the swarming inhibition phenotype. The authors should acknowledge this fact and discuss how it may affect their results and interpretation.

- As the only difference is the presence of rsmQ, Fig 6C could be simplified by changing the plasmid names (pME6032 and prsmQ) to pME6032 and pME6032-rsmQ.

- Figure 7c. The number of technical and biological replicates should be indicated.

- L341-343. Please provide statistical support for this claim.

- L349-350. This sentence provides much-needed context for the carbon source experiments in figures 7 and S3. Please consider providing this information to the reader before moving the experiments to different carbon sources.

- L377-382. The biphasic effect warrants more investigation. It makes me wonder whether the results of the biolog assays are also subject to a stochastic behavior and, therefore the conclusions drawn from the experiment (L358-370). Would selecting a different colony change the observed results?

- The discussion section is overly too long. Please reduce its extension thoroughly by eliminating redundancy with the results section.

- L409. PROKKA database? COMPASS instead?

- The authors only found an effect on conjugation rate for one out 6 conditions tested (3 carbon sources in liquid and solid conditions). I think this is a bit too anecdotal to draw general conclusions like the ones in lines 435-443. Please tone down and add a bit of nuance.

---

## [Editor Report · Decision Letter 2]

13 Dec 2022

Dear Dr. Malone,

Thank you for your patience while we considered your revised manuscript "Plasmid manipulation of bacterial behaviour through translational regulatory crosstalk" for publication as a Research Article at PLOS Biology. This revised version of your manuscript has been evaluated by the PLOS Biology editors, and the Academic Editor.

Based on our Academic Editor's assessment of your revision, we are likely to accept this manuscript for publication, provided you satisfactorily address the following data and other policy-related requests.

1. DATA POLICY:

A) Supplementary files (e.g., excel). Please ensure that all data files are uploaded as 'Supporting Information' and are invariably referred to (in the manuscript, figure legends, and the Description field when uploading your files) using the following format verbatim: S1 Data, S2 Data, etc. Multiple panels of a single or even several figures can be included as multiple sheets in one excel file that is saved using exactly the following convention: S1_Data.xlsx (using an underscore).

B) Deposition in a publicly available repository. Please also provide the accession code or a reviewer link so that we may view your data before publication.

Regardless of the method selected, please ensure that you provide the individual numerical values that underlie the summary data displayed in the following figure panels as they are essential for readers to assess your analysis and to reproduce it: Figures 1ABC, 2AB, 3AB, 4, 5AC, 6C, 7ABC, 8AB, and supplementary figures S1ABC, S2ABC, S3ABCD, S5ABC, S7, S8, S9AB, S10, S11ABCD.

**Please also ensure that figure legends in your manuscript include information on where the underlying data can be found, and ensure your supplemental data file/s has a legend.**

2. We suggest to change the title to an active voice: "Plasmids manipulate bacterial behaviour through translational regulatory crosstalk".

We expect to receive your revised manuscript within two weeks.

*Published Peer Review History*

*Press*

Sincerely,

Paula

---

Senior Editor,

pjaureguionieva@plos.org,

PLOS Biology

---

## [Editor Report · Decision Letter 3]

4 Jan 2023

Dear Dr. Malone,

Thank you for the submission of your revised Research Article "Plasmids manipulate bacterial behaviour through translational regulatory crosstalk" for publication in PLOS Biology. On behalf of my colleagues and the Academic Editor, Lotte Søgaard-Andersen, I am pleased to say that we can in principle accept your manuscript for publication, provided you address any remaining formatting and reporting issues. These will be detailed in an email you should receive within 2-3 business days from our colleagues in the journal operations team; no action is required from you until then. Please note that we will not be able to formally accept your manuscript and schedule it for publication until you have completed any requested changes.

PRESS

Sincerely, 

Paula 

---

Senior Editor

PLOS Biology
